# ADVERSARIAL MIXUP UNLEARNING

**Zhuoyi Peng**    **Yixuan Tang**    **Yi Yang**
Department of ISOM, The Hong Kong University of Science and Technology
`{zpengaf, ytangch}@connect.ust.hk, yiyang@ust.hk`

## ABSTRACT

Machine unlearning is a critical area of research aimed at safeguarding data privacy by enabling the removal of sensitive information from machine learning models. One unique challenge in this field is catastrophic unlearning, where erasing specific data from a well-trained model unintentionally removes essential knowledge, causing the model to deviate significantly from a retrained one. To address this, we introduce a novel approach that regularizes the unlearning process by utilizing synthesized mixup samples, which simulate the data susceptible to catastrophic effects. At the core of our approach is a generator-unlearner framework, MixUnlearn, where a generator adversarially produces challenging mixup examples, and the unlearner effectively forgets target information based on these synthesized data. Specifically, we first introduce a novel contrastive objective to train the generator in an adversarial direction: generating examples that prompt the unlearner to reveal information that should be forgotten, while losing essential knowledge. Then the unlearner, guided by two other contrastive loss terms, processes the synthesized and real data jointly to ensure accurate unlearning without losing critical knowledge, overcoming catastrophic effects. Extensive evaluations across benchmark datasets demonstrate that our method significantly outperforms state-of-the-art approaches, offering a robust solution to machine unlearning. This work not only deepens understanding of unlearning mechanisms but also lays the foundation for effective machine unlearning with mixup augmentation.

## 1 INTRODUCTION

Machine unlearning (Bourtoule et al., 2021) has gained significant attention due to growing concerns about data privacy. In particular, legislators have enacted regulations such as the GDPR, which grants data owners the "right to be forgotten". This has spurred the development of machine unlearning techniques that enable models to forget sensitive data. While retraining models without sensitive data can achieve accurate unlearning, it is often impractical due to the substantial computational costs involved. As a result, researchers and practitioners are increasingly exploring *approximate unlearning* methods (Thudi et al., 2022; Chen et al., 2023; Chundawat et al., 2023a; Kurmanji et al., 2024; Shen et al., 2024). These aim to create models that function as if they were retrained from scratch, but without the prohibitive expenses associated with full retraining.

While approximate unlearning holds promise, it faces the issue of catastrophic unlearning—a phenomenon where a target unlearner model, during the process of unlearning certain data, inadvertently forgets or loses knowledge it should retain, resulting in a divergence from the retrained one. Although recent advances (Chen et al., 2023; Chundawat et al., 2023a; Shen et al., 2024) have introduced retention operations on the remaining data to facilitate preserving the model's generalizability, the issue of catastrophic unlearning persists. Despite its importance, effective strategies to address this problem remain inadequately understood (Choi et al., 2024). To bridge this research gap, our paper answers the key question: *How can we overcome catastrophic unlearning to ensure precise forgetting—erasing target information without compromising essential knowledge?*

To understand why the retaining process *cannot* handle catastrophic effects, we provide a toy example in Figure 1, which shows the challenge of unlearning a specific class for a classifier. This example demonstrates how issues arise when the forgetting (applied to the *Forgetting* samples) and the retaining (applied to the *Remaining* samples) potentially interfere with one another. Specifically, the forgetting process may disrupt the retaining process, weakening the model's ability to general-

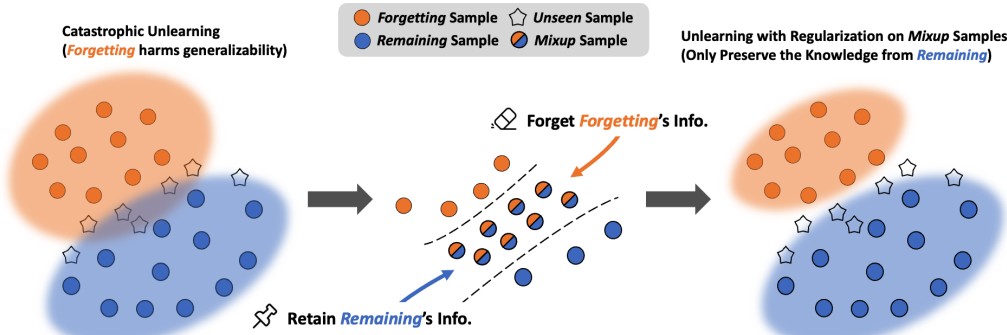

Figure 1: A toy example of unlearning a focal class (in orange) to motivate our solution. The shaded orange represents the effects of forgetting, while the shaded blue indicates the retention of remaining knowledge. This example demonstrates how forgetting can inadvertently harm the knowledge that should be retained, as shown by the overlapping shaded regions, ultimately degrading the generalizability on unseen data (shown as stars). Notably, we can sythesize mixup samples—which are strategically mixed from *Forgetting* and *Remaining*—to mimic the data vulnerable to the overlapping catastrophic effects. By removing the forgetting effects and retaining remaining knowledge on these mixup samples, we can overcome catastrophic unlearning.

ize—especially in the intermediate space between the *Forgetting* and *Remaining* samples. However, by strategically using mixup (Zhang et al., 2018; Liu et al., 2022; Qin et al., 2024), which generates intermediate samples through linear interpolation between the forgetting and remaining data, we can address this issue. These synthesized mixup samples can simulate data points with catastrophic effects, enabling us to regularize the target unlearner for more effective unlearning—by minimizing the negative forgetting impact on the mixup samples, we make the model behave as if it were trained solely on the remaining data.

To fully leverage mixed samples for addressing catastrophic unlearning, we propose a novel generator-unlearner framework going beyond simply adopting the vanilla mixup: an adversarial generator creates challenging mixup samples from mixing *Forgetting* and *Remaining* data, and then the unlearner is regularized with these hard samples to enhance unlearning robustness. Specifically, unlike traditional mixup relying on handcrafted rules, our method incorporates a mixup generator trained by a proposed contrastive loss in an adversarial manner. This generator is designed to synthesize mixed samples that deliberately challenge the unlearner, causing it to forget remaining knowledge while revealing information about forgetting data (a reversed direction of unlearning which removes *Forgetting* and retains *Remaining*). The target unlearner then processes both these synthetic mixed samples and real data, employing two distinct contrastive losses—one for each data type—to effectively forget concerning information while retaining essential knowledge. Notably, the overall method can perform unlearning without explicit labels for forgetting and remaining data, making it particularly suitable for scenarios where models are initially trained on sparsely-annotated datasets and labels are largely absent during unlearning. Extensive experiments across benchmark datasets demonstrate that our method significantly outperforms existing state-of-the-art unlearning techniques (Bourtoule et al., 2021; Chen et al., 2023; Chundawat et al., 2023a; Kurmanji et al., 2024; Choi et al., 2024; Shen et al., 2024), across both label-agnostic and label-aware setups.[1] Our work underscores the potential benefits of leveraging mixup samples for machine unlearning.

The contributions are as follows: First, we introduce a novel unlearning approach that leverages mixup samples to regularize the unlearner, offering a new strategy for addressing the issue of catastrophic unlearning. Second, we extend beyond the traditional mixup by proposing a generator-unlearner framework, where adversarially-generated examples improve the unlearning process. This framework is trained using novel contrastive losses without any need for explicit labels for forgetting and retaining data. Third, empirical evaluations across multiple benchmark datasets demonstrate that our approach surpasses existing unlearning techniques. We have released the code for MixUnlearn.[2]

---

[1]*Label-agnostic unlearning* refers to algorithms that facilitate unlearning without any need for labels, which is particularly advantageous in real-world scenarios where much data may be unannotated, as is often the case in weakly labeled or semi-supervised learning environments. In contrast, *label-aware unlearning* methods depend on labeled data to execute the unlearning process.

[2]https://github.com/realjeremybot/MixUnlearn.git

## 2 RELATED WORK

**General Unlearning Work**. Unlearning techniques are generally divided into two categories: exact unlearning and approximate unlearning. Exact unlearning focuses on efficiently retraining a model using *only* the remaining data. For example, the SISA algorithm (Bourtoule et al., 2021) enhances retraining efficiency by initially training multiple model checkpoints on distinct data shards, only retraining the specific checkpoints linked to the data that must be forgotten. As models and datasets become increasingly complex, the limitations of exact unlearning have led to the rise of approximate unlearning methods. These methods aim to modify an already trained model by utilizing both forgotten and retained data, striving to replicate the performance of a model retrained from scratch (Nguyen et al., 2020; Golatkar et al., 2020b;a; Liu et al., 2021; Thudi et al., 2022; Chen et al., 2023; Tarun et al., 2023; Chundawat et al., 2023a; Kurmanji et al., 2024). Noteworthy recent advancements include Chundawat et al. (2023a), which employs two teacher models—one teacher helps the unlearner retain essential knowledge, while the other guides forgetting undesired information; Chen et al. (2023), which refines decision boundaries to facilitate unlearning; and Thudi et al. (2022), which reverses parameter updates associated with the *Forgetting* data.

**Data Augmentation for Unlearning**. Recent research has explored the use of data augmentation techniques to support the unlearning process (Chundawat et al., 2023b; Tarun et al., 2023; Huang et al., 2021; Choi et al., 2024). Notable methods include UNSIR (Tarun et al., 2023), GLI (Choi et al., 2024), and DSMixup (Zhou et al., 2022). UNSIR generates artificial noise to increase classification loss, thereby facilitating unlearning with noisy data. GLI perturbs retained samples with noise and maintains model generalizability by preserving performance on noisy retained samples. Our method offers two advantages over these approaches: (1) Unlike UNSIR, which is limited to class-level unlearning, our approach is more versatile, supporting both class-level and data-level unlearning; and (2) While GLI attempts to preserve utility through sample perturbation, it does not adequately address catastrophic unlearning. DSMixup, one notable method, improves the SISA framework by converting retraining shards into a smaller set of mixup shards, enhancing retraining efficiency. Although DSMixup employs mixup, its purpose and adaptation differ largely from ours. DSMixup is aimed at "exact unlearning" and improving retraining efficiency, while our approach focuses on mitigating catastrophic effects in well-trained models, as an "approximate unlearning" method. Additionally, DSMixup mixes data shards, whereas our method mixes *Forgetting* and *Remaining* samples. While DSMixup prioritizes efficiency, sometimes at the expense of accuracy, our method can preserve model generalizability. To our knowledge, we are among the first to leverage mixup to address catastrophic unlearning, a challenge unique to approximate unlearning.

**Mixup for Traditional Machine Learning**. Mixup techniques have been widely used in both supervised and semi-supervised learning settings (Zhang et al., 2018; Verma et al., 2022; Berthelot et al., 2019; Jin et al., 2024). Notable advanced methods include AutoMix (Liu et al., 2022) and AdAutoMix (Qin et al., 2024), which employ learnable generators to create mixup samples, though their optimization objectives differ. Despite their success in traditional learning applications, the use of mixup strategies in the context of unlearning remains underexplored. In particular, training a mixup generator to effectively enhance the unlearning process is still an open challenge.

## 3 PRELIMINARY

Let $f_D$ be a deep model trained on dataset $D$, which maps instance $x \in \mathcal{X}$ to a distribution $y \in \mathcal{Y}$ over class labels. Unlearning on $f_D$ is to eliminate the knowledge associated with a subset of the data, denoted as $D_f \subset D$, while retaining the knowledge derived from the remaining data, $D_r = D \setminus D_f$. With an *approximate unlearning* approach, denoted by $U$, the original model $f_D$ is transformed into an unlearned model $f_U = U(f_D, D_r, D_f)$. The goal of the unlearned model $f_U$ is to approximate the performance of $f_{D_r}$, a model trained solely on $D_r$. We refer to $f_D$ as the initial model, $f_U$ as the unlearner, $D_f$ as the *Forgetting* dataset, and $D_r$ as the *Remaining* dataset.

Conceptually, the unlearning algorithm $U$ can be categorized into two types: label-agnostic and label-aware. Label-agnostic unlearning does not require access to labels, making it particularly suited for scenarios where the initial model is trained on largely unannotated datasets, and labels are not guaranteed to be available during the unlearning phase. In contrast, label-aware unlearning relies on label information during the unlearning process to facilitate removal of knowledge.

Figure 2: The goal of our generator is to produce hard mixed samples that challenge the unlearner. These mixed samples can prompt the unlearner to reveal information related to *Forgetting* and simultaneously disrupt the retention of *Remaining*, reversing the direction of unlearning process (the goal of unlearning is to forget *Forgetting* while preserving *Remaining*). To train the generator, we hold the unlearner fixed and update the generator's parameters with a proposed contrastive objective of Eq. 3. The unlearner then performs effective unlearning based on these mixed samples.

# 4 MixUnlearn: An Adversarial Mixup Unlearning Framework

Why does retention apply to the remaining data, yet catastrophic effects on generalizability persist? This issue likely arises from insufficient model regularization in intermediate regions or interpolation zones, where the effects of forgetting and retention intersect. For example, consider a classifier trained to recognize cats and dogs, where we aim for the model to forget the cat class while retaining knowledge of the dog class. Although retention works for training samples of dogs, unseen dog samples that exist in the overlapping or interpolation space between cats and dogs gain minimal benefit. These unseen samples are likely close to the cat class in feature space, making it challenging to fully mitigate the forgetting effect. Consequently, the model's ability to generalize to these unseen dog samples diminishes due to the residual impact of forgetting, leading to catastrophic unlearning.

To mitigate catastrophic unlearning, we propose integrating mixup samples into the machine unlearning process. Mixup, introduced by Zhang et al. (2018), is a data augmentation technique that creates new samples by linearly interpolating between two existing samples. By strategically mixing samples from *Forgetting* and *Remaining*, we simulate instances that experience the conflicting forces of forgetting and retention. These mixup samples allow us to regularize the unlearning process, reducing the catastrophic effects associated with them.

However, vanilla mixup may *not* pose a significant challenge to the unlearning process, as it relies on fixed patterns or linear combinations that the unlearner can handle with relative ease. To more effectively utilize mixed samples, we propose a mixup generator that adversarially produces more challenging mixup data, optimized through a novel contrastive loss. These samples are specifically crafted to exploit the weaknesses of the unlearner by pushing it to discard the *Remaining* knowledge while revealing information about *Forgetting*—the reverse process of unlearning, whose goal is to forget the *Forgetting* and retain the *Remaining*. By introducing these challenging samples, the unlearner is exposed to complex scenarios, resulting in stronger regularization compared to standard mixup. Next, we will introduce our adversarial generator and then illustrate how to incorporate these challenging mixup samples into unlearning.

## 4.1 Learning Adversarial Generator to Challenge Unlearner

The illustration of our generator is shown in Figure 2. In the generator's forward pass, each mixed sample is mixed from one instance in *Forgetting* set with one in *Remaining* set. Unlike standard mixup, which uses simple interpolation (i.e., $x_{ij}^{mix} = \lambda x_i + (1 - \lambda)x_j$), our generator employs a learnable mixing function $g$. This allows for tailoring loss function to increase the complexity of the mixed samples to challenge the unlearner. The mixed sample $x_{ij}^{mix}$ is defined as:

$$x_{ij}^{mix} = g(x_i, x_j, \lambda), \tag{1}$$

where $g$ is the learnable mixing function, $x_i$ is a sample from *Forgetting* set, $x_j$ is from *Remaining* set, and $\lambda$, drawn from a Beta distribution, determines the mixing ratio between $x_i$ and $x_j$.

**Mixing Forgetting and Remaining**. To operationalize a learnable generator, we utilize the parameterized network module $MixBlock$ from prior work (Qin et al., 2024) as the generator $g$. This $MixBlock$ can generate attention scores for each element of $x_i$ and $x_j$, allowing for dynamic mixing of the two samples. Since $MixBlock$ is trainable, we can also tailor the generator's behavior with a customized loss function. The main distinctions of our generator compared to previous work (Qin et al., 2024) in traditional learning are: 1) its novel application within the emerging domain of

machine unlearning; 2) a strategic mixup of *Forgetting* and *Remaining* samples, rather than random mixing; and 3) the introduction of a new contrastive objective that efficiently directs the generator to challenge the unlearner, as we will discuss in detail later.

Specifically, with learnable $MixBlock$ module to produce mixup samples, we extend $g$ as:

$$x_{ij}^{mix} = MixBlock(h_D(x_i), h_D(x_j), \lambda), \tag{2}$$

where $h_D(x_i)$ and $h_D(x_j)$ are dense feature representations obtained from initial model $f_D$.[3] Notably, $MixBlock$ has only 66K parameters and efficient to train.[4]

**Adversarial Optimization**. The distinctiveness of our mixup examples lie in their ability to challenge the unlearning process, where they force the unlearner to expose information about what is being forgotten while leading it to lose knowledge that should be retained (this is an reversed direction of unlearning). This adversarial goal is achieved by optimizing the generator to output hard examples with a proposed contrastive loss. Specifically, a novel contrastive loss is applied on mixed sample $x_{ij}^{mix}$ to penalize the generator $g$ to generate hard sample (note the "-" sign to achieve adversarial purpose) for challenging unlearner $f_U$:

$$L_{\text{gen}} = - \sum_{x_j \in B_r} \log \left( \frac{\exp\left((1-\lambda) \cdot SimLoss(f_U(x_{ij}^{mix}), p(x_j))\right)}{\sum_{x_i \in B_f} \exp\left(\lambda \cdot SimLoss(f_U(x_{ij}^{mix}), p(x_i))/\tau_{gen}\right)} \right), \tag{3}$$

with $p(\cdot)$ operation retrieves the one-hot label or "sharpened" (Goodfellow et al., 2016) class distribution generated by initial model $f_D$:[5]

$$p(x) = \begin{cases} y & \text{if the label } y \text{ of } x \text{ is available (label-aware)} \\ \text{Sharpen}(f_D(x)) & \text{otherwise (label-agnostic)} \end{cases}. \tag{4}$$

The intuition behind the contrastive loss in Eq. 3 operates on two key principles. First, it encourages the model to reveal the distributional information of the forgetting sample (*i.e.*, $p(x_i)$) from the output of the mixed sample ($f_U(x_{ij}^{mix})$). This is reflected in the denominator, where the negative sign serves an adversarial purpose. Second, it simultaneously disrupts the retention of distributional knowledge about the retaining sample ($p(x_j)$) from the output of the mixed sample (again, $f_U(x_{ij}^{mix})$). This is indicated by the numerator, with a negative sign.[6] Together, this loss essentially reverses the unlearning process: while "the objective of unlearning" is to forget the forgetting sample and retain knowledge of the remaining sample, this loss works in the opposite direction.

Specifically, in Eq. 3, $B_f$ denotes a batch of data to be forgotten, while $B_r$ represents a batch of data to be retained. $x_i$ refers to a sample to be forgotten, $x_j$ to a sample to be retained, and their mixed sample is given by $x_{ij}^{mix} = g(x_i, x_j, \lambda)$. The $\lambda$ controls the weights between two $SimLoss$ for $x_i$ and $x_j$ according to mixing ratio. $SimLoss$ is defined as the cosine similarity loss, expressed as $(1 - \text{cosine similarity})$. Moreover, the hyperparameter $\tau_{\text{gen}}$ adjusts the sensitivity of $SimLoss$, particularly in the denominator.

## 4.2 Unlearning with Adversarial Mixed Samples

Recall that we have obtained the mixed sample $x_{ij}^{mix}$, which encapsulates information from *Forgetting* and *Remaining* and is designed to challenge the unlearner. We then feed $x_{ij}^{mix}$ into the unlearner

---

[3] Machine unlearning starts with a well-trained $f_D$, allowing us to use its existing feature extractor.

[4] $MixBlock$ uses parameterized convolutional networks to extract deeper features and compute elementwise attention masks $M_i$ to mask the elements in $x_i$. With both $x_i$ and $x_j$ masked, the mixed sample is computed as $x_{ij}^{mix} = x_i \odot M_i + x_j \odot (\mathbf{1} - M_i)$, where $\mathbf{1}$ is a tensor with the same dimensions as $x_i$, with all elements set to 1. Full details on the $MixBlock$ module can be found in Section 3.2 of Qin et al. (2024).

[5] The formula of $Sharpen$ is shown in Appendix A.12. The $Sharpen$ results in sharper contrasts during optimization, ensuring that the generated examples are more challenging to increase adversarial difficulty.

[6] Although other adversarial objectives were explored, we found this contrastive loss perform the best. This success is attributed to the appropriate application of $SimLoss$, which effectively bounds the loss terms and reduces the risk of exposing the model to large negative gradients that could negatively impact its parameters.

$f_U$, ensuring that the unlearner's prediction no longer retains knowledge of $x_i$, while only preserving that of $x_j$. The regularization on unlearner with mixed sample is achieved by optimizing $f_U$ with:

$$L_{\text{mix}} = \sum_{x_j \in B_r} \log \left( \frac{\exp\left((1-\lambda) \cdot SimLoss(f_U(x_{ij}^{mix}), p(x_j))\right)}{\sum_{x_i \in B_f} \exp\left(\lambda \cdot SimLoss(f_U(x_{ij}^{mix}), p(x_i))/\tau_{mix}\right)} \right), \tag{5}$$

This loss function only change Eq. 3 with different sign and works in reverse: it directs $f_U$ to remove information about $x_i$ while retaining the knowledge of $x_j$. A temperature hyperparameter $\tau_{mix}$, again, is used to adjust the sensitivity of the similarity loss.

To further enhance the unlearning process, we add another contrastive loss that operates on the original real samples $x_i$ and $x_j$:

$$L_{\text{real}} = \sum_{x_j \in B_r} \log \left( \frac{\exp\left(SimLoss(f_U(x_j), p(x_j))\right)}{\sum_{x_i \in B_f} \exp\left(SimLoss(f_U(x_i), p(x_i))/\tau_{real}\right)} \right). \tag{6}$$

This loss helps the model unlearn on original real examples $x_i$ and $x_j$, reinforcing the retention of $f_D(x_j)$ while forgetting distributional information $f_D(x_i)$ ($x_i$ is a forgetting sample and $x_j$ is a remaining sample). Finally, we combine the two losses in a weighted manner to optimize the unlearner $f_U$:

$$L_{\text{unlearn}} = L_{\text{mix}} + \omega L_{\text{real}}, \tag{7}$$

where $\omega$ balances the importance of the two losses. Our overall framework operates by iteratively optimizing generator $g$ and unlearner $f_U$. Notably, $g$ only needs to learn 66K parameters in the lightweight $MixBlock$, a significantly lower number compared to the unlearner (*e.g.*, 11.3M parameters in ResNet-18). This design contributes to the overall efficiency. To further enhance efficiency, we optimize $g$ at a specific interval during adversarial training, rather than at every unlearning iteration. For example, in our experiments, we optimize $g$ once every four iterations.

## 5 EXPERIMENTS

We conduct a series of experiments to validate the unlearning effectiveness of MixUnlearn.

### 5.1 DATASETS AND MODELS

Following prior works (Bourtoule et al., 2021; Chen et al., 2023; Shen et al., 2024), we conduct experiments on four datasets: CIFAR10 (Krizhevsky et al., 2009), SVHN (Netzer et al., 2011) and MNIST (LeCun et al., 1998), FASHION-MNIST (Xiao et al., 2017). For the CIFAR10 and SVHN, we adopt an 18-layer ResNet architecture (He et al., 2016). For the MNIST and FASHION, we use a simple convolutional neural network (CNN) (LeCun et al., 1995) with two convolutional layers.

### 5.2 BASELINES

We employ two categories of baselines for comparison: label-aware, which require full label information, and label-agnostic, which do not rely on any explicit label to function unlearning.

**Label-Aware Baselines**. This category includes Retrain, the gold standard. The baselines also encompass Boundary (Chen et al., 2023), SISA (Bourtoule et al., 2021), Unroll (Thudi et al., 2022), T-S (Chundawat et al., 2023a), SCRUB (Kurmanji et al., 2024), GLI (Choi et al., 2024), DSMixup (Zhou et al., 2022), and Label-Agnostic Forgetting with Repairing (LAF+R) (Shen et al., 2024). Among these, LAF+R stands out by performing unlearning at the representation level using multiple VAEs (Kingma & Welling, 2014), followed by a single retraining epoch. Another notable method is GLI, a recent data-augmentation-based approach that applies augmentation to retained samples to preserve generalizability. Although DSMixup employs mixup, it serves a different goal: it mixes data shards into a smaller set of mixed shards to accelerate retraining, whereas our approach mixes *Forgetting* and *Remaining* to maintain generalizability. Additionally, DSMixup is designed for exact

unlearning, whereas our method is tailored for approximate unlearning. Full label information is provided to both these baselines and MixUnlearn during our comparisons.

**Label-Agnostic Baselines**. This category includes Label Agnostic Forgetting (LAF) (Shen et al., 2024), a pioneering approach. Due to the scarcity of label-agnostic baselines, we propose two baselines. The first, RandLabel, assigns random labels to the forgetting data using a MeanSquaredError (MSE) loss. An additional MSE loss is applied to preserve the original output distribution for the remaining data. The second, L-Mix, extends the LAF by incorporating mixup. In L-Mix, mixed samples are created by mixing forgetting data with retaining data using the standard rule. The labels for these mixed samples are mixed by random labels (for the forgetting) and the model's self-generated labels (for the retaining). An MSE loss is applied to these mixed samples and labels upon LAF. The L-Mix is intended to assess the potential of a straightforward adaptation of mixup. No label information is provided to these baselines and MixUnlearn during our comparisons.

Full details are shown in Appendix A.3. Experiments are repeated five times with random seeds.

## 5.3 EVALUATION SETUP AND METRICS

We evaluate MixUnlearn using unlearning setups following Shen et al. (2024). First, in the **Class-Level Unlearning** setup, all data from class 0 is removed. We obtain the initial model for unlearning by training with full dataset with labels. Then we get the Retrain by retraining on the labeled *Remaining* subset, following Shen et al. (2024). Second, in the **Data-Level Unlearning (Basic)** setup, we randomly remove 40% of the training data labeled with classes 5 through 9. The model initialization and retraining process follow the same procedure as in the Class-Level Unlearning.

For robustness check, we introduce two additional configurations, extending the Data-Level Unlearning (Basic) setup. The first involves assessing unlearning methods in a noisy label scenario, referred to as **Data-Level Unlearning (Noisy)**. The second evaluates unlearning in a semi-supervised setting, termed **Data-Level Unlearning (Semi-Supervised)**. The key findings are presented in Section 5.8, with detailed descriptions provided in Appendix A.6.

**Metrics**. In the Class-Level Unlearning setup, the evaluation metrics are: **Test**$_r$ (test accuracy on the remaining classes), **Test**$_f$ (test accuracy on the forgotten class), and **ASR** (attack success rate of membership inference attacks, as proposed by Shokri et al. (2017)). For the Data-Level Unlearning setups (Basic, Noisy, and Semi-Supervised), the relevant metrics include: **Train**$_r$ (accuracy on the remaining training data post-unlearning), **Train**$_f$ (accuracy on the training data targeted for removal), and **Test** (test accuracy on the test dataset). These metrics reflect the performance, with results closer to Retrain indicating more effective unlearning outcomes.

## 5.4 MAIN RESULTS

We present the main results upon two unlearning scenarios: class-level and data-level (Basic), along with two label configurations: agnostic and aware. For MixUnlearn, depending on label awareness, we apply different forms of Eq. 4, resulting in two variants for label-agnostic and label-aware setups.

First, Tables 1 and 2 show that our method consistently outperforms existing baselines across a range of configurations, including both label-agnostic and label-aware settings, as well as data-level and class-level unlearning (while achieving comparable results in MNIST data-level unlearning). Key metrics, such as test accuracy and attack success rate (ASR), show that our method delivers competitive results close to Retrain. For example, in class-level unlearning tests on CIFAR-10 and SVHN, our label-aware approach achieves average test accuracies of 87.10% and 93.95%, respectively, outperforming state-of-the-art methods such as LAF+R, SISA (an exact unlearning approach), GLI (a recent data augmentation-based method for approximate unlearning), and DSMixup (an exact unlearning method utilizing mixup to accelerate retraining). These findings highlight the effectiveness of our unlearning strategy. We visualize a mixed sample in Appendix A.9, and more results on ImageNet (Deng et al., 2009) with ViT (Dosovitskiy et al., 2021) are shown in Section A.11.

Second, although our proposed baseline, L-Mix, outperforms LAF by incorporating standard mixup, our approach takes it a step further by fully utilizing mixup with hard examples and incorporating a series of effective contrastive losses. While simpler mixup implementations do lead to performance improvements, our method achieves significantly greater gains. For instance, MixUnlearn improves

upon L-Mix on CIFAR-10 in Class-Level Unlearning (86.32 vs. 82.34 in $\text{Test}_r$). This highlights that our method is not only highly effective but also innovative in how it leverages mixup.

Third, it is crucial to recognize that the label-agnostic setup poses significantly greater challenges compared to its label-aware counterpart. The $\text{Test}_r$ metric for label-agnostic MixUnlearn clearly highlights this disparity when compared to the label-aware version, underscoring the challenges unlearning algorithms face without explicit labels. Nevertheless, our method effectively handles both label-agnostic and label-aware setups.

Table 1: Class-Level Unlearning Performance (Mean%±Std%). Closer to "Retrain" is better. "Aware" is label-aware; "Agnostic" is label-agnostic. Bold indicates the best result for each metric.

| | | CIFAR-10 | | | | SVHN | | |
|---|---|---|---|---|---|---|---|---|
| | Method | $\text{Test}_r$ | $\text{Test}_f$ | ASR | Method | $\text{Test}_r$ | $\text{Test}_f$ | ASR |
| **Aware** | Retrain | 86.80±0.89 | 0±0 | 67.98±2.21 | Retrain | 94.20±0.78 | 0±0 | 59.63±1.77 |
| | NegGrad | 58.48±3.41 | 0±0 | 51.53±1.21 | NegGrad | 77.12±1.56 | 5.98±1.33 | 53.22±2.10 |
| | Boundary | 82.98±1.98 | 1.52±0.33 | 62.33±2.98 | Boundary | 91.28±1.02 | 13.12±3.91 | 61.34±3.05 |
| | SISA | 73.01±0.56 | 0±0 | 51.53±0.12 | SISA | 92.04±0.81 | 0±0 | 62.45±1.34 |
| | Unrolling | 83.89±2.02 | 0±0 | 67.32±1.92 | Unrolling | 92.01±1.19 | 88.12±3.12 | 58.30±3.02 |
| | T-S | 86.31±1.12 | 5.20±1.32 | 46.98±3.26 | T-S | 92.89±0.82 | 7.86±2.06 | 48.97±0.88 |
| | SCRUB | 34.12±1.23 | 0±0 | 49.89±1.23 | SCRUB | 20.33±0.65 | 0±0 | 64.21±1.12 |
| | DSMixup | 64.31±2.01 | 0±0 | 50.83±1.93 | DSMixup | 82.31±1.90 | 0±0 | 50.62±1.77 |
| | GLI | 84.82±0.65 | 47.28±5.70 | 73.03±0.80 | GLI | 93.44±0.50 | 63.22±7.77 | 86.10±0.32 |
| | LAF+R | 87.20±0.69 | 0.20±0.04 | 56.89±1.23 | LAF+R | 91.35±0.73 | 0±0 | 61.89±1.37 |
| | **Ours** | **87.10±0.78** | **0±0** | **68.30±2.77** | **Ours** | **93.95±0.69** | **0±0** | **59.97±2.68** |
| **Agnostic** | LAF | 82.01±0.89 | 3.10±0.98 | 51.34±1.27 | LAF | 84.89±1.34 | 0.78±0.22 | 56.45±0.65 |
| | RandLabel | 81.60±0.57 | 19.90±0.16 | 64.76±1.56 | RandLabel | 91.05±1.90 | 91.11±1.55 | 62.59±0.83 |
| | L-Mix | 82.34±0.99 | 0.66±1.57 | 53.61±1.88 | L-Mix | 88.74±0.88 | 0.30±0.47 | 52.00±1.11 |
| | **Ours** | **86.32±0.56** | **0±0** | **68.48±0.67** | **Ours** | **93.40±1.35** | **0±0** | **62.18±0.72** |

| | | MNIST | | | | FASHION-MNIST | | |
|---|---|---|---|---|---|---|---|---|
| | Method | $\text{Test}_r$ | $\text{Test}_f$ | ASR | Method | $\text{Test}_r$ | $\text{Test}_f$ | ASR |
| **Aware** | Retrain | 98.79±0.20 | 0±0 | 26.56±1.75 | Retrain | 92.71±0.47 | 0±0 | 38.04±2.11 |
| | NegGrad | 98.89±0.22 | 81.89±5.23 | 38.12±2.11 | NegGrad | 88.91±0.92 | 1.23±0.34 | 37.98±2.45 |
| | Boundary | 98.43±0.37 | 96.24±1.32 | 38.89±2.41 | Boundary | 86.41±1.78 | 1.32±0.38 | 38.45±2.13 |
| | SISA | 99.11±0.05 | 0±0 | 50.24±0.35 | SISA | 92.33±0.12 | 0±0 | 49.80±0.13 |
| | Unrolling | 97.32±0.76 | 83.41±5.41 | 37.45±4.12 | Unrolling | 87.41±1.23 | 0.40±0.12 | 41.41±2.13 |
| | T-S | 61.53±10.45 | 0.21±0.05 | 36.83±3.12 | T-S | 91.51±0.64 | 21.43±5.55 | 26.13±0.78 |
| | SCRUB | 99.12±0.04 | 89.31±2.13 | 32.41±3.21 | SCRUB | 91.32±0.64 | 0.51±0.13 | 35.14±0.89 |
| | DSMixup | 99.09±0.18 | 0±0 | 25.01±0.59 | DSMixup | 91.56±0.47 | 0±0 | 35.33±1.01 |
| | GLI | 98.99±0.10 | 99.51±0.32 | 61.37±0.22 | GLI | 90.87±0.61 | 86.42±3.74 | 60.93±0.04 |
| | LAF+R | 99.12±0.05 | 0.23±0.05 | 25.01±1.24 | LAF+R | 91.85±0.34 | 0.33±0.07 | 34.89±2.89 |
| | **Ours** | **98.85±0.04** | **0±0** | **27.13±0.77** | **Ours** | **92.82±0.66** | **0±0** | **38.15±2.54** |
| **Agnostic** | LAF | 97.97±0.26 | 0.27±0.06 | 49.31±3.23 | LAF | 89.88±0.80 | 3.12±0.82 | 31.89±1.32 |
| | RandLabel | 95.68±1.21 | 50.91±10.88 | 27.89±2.12 | RandLabel | 87.96±1.14 | 39.1±3.36 | 33.29±0.85 |
| | L-Mix | 98.05±0.14 | 0±0 | 24.01±0.19 | L-Mix | 90.15±1.33 | 5.01±2.94 | 32.95±1.59 |
| | **Ours** | **98.88±0.07** | **0±0** | **27.37±1.60** | **Ours** | **91.80±1.33** | **0±0** | **38.07±0.65** |

## 5.5 ABLATION

We systematically remove individual components from our method (label-agnostic version), with the results in Table 3 and Table 4 (Appendix A.4). Specifically, in the *w/o* MB ablation, we replace the hard mixup samples with vanilla mixup samples, controlling mix ratio with different values of $\alpha$: (1) $\alpha = 0.35$, which produces mixup ratios skewed close to 1 (the extreme case of 1 represents no mixing); (2) $\alpha = 1.5$, which generates mixup ratios centered near 0.5 (indicating equal mixing); and (3) $\alpha = 0.75$, which produces a broader range of ratios. Notably, this ablation differs from the proposed L-Mix baseline in main results, as it uses our contrastive losses (Eq. 5 and Eq. 6).

The results for *w/o* MB highlight the advantages of using a learnable generator, which improves both test accuracy and ASR. Among the configurations, $\alpha = 0.75$ consistently yields better results, likely due to the creation of more diverse mixup samples, in contrast to $\alpha = 0.35$ and $\alpha = 1.5$, which tend to constrain the mixup ratios closer to 1 or 0.5, respectively. Additionally, the combination of $L_{\text{mix}}$ and $L_{\text{real}}$ performs better together than individually. Finally, the sharpening mechanism provides a modest improvement. Similar trends appear in the data-level unlearning results (Appendix Section A.4). For example, in the FASHION-MNIST data-level unlearning setup, the inclusion of MixBlock results in effective forgetting of samples, achieving an average $\text{Train}_f$ (the accuracy on forgetting data) of 91.99, which closely matches the 92.15 achieved by the Retrain. In contrast, the configura-

Table 2: Data-Level (Basic) Unlearning Performance (Mean%±Std%).

| | Method | Train$_r$ | Train$_f$ | Test | ASR | Method | Train$_r$ | Train$_f$ | Test | ASR |
|---|---|---|---|---|---|---|---|---|---|---|
| | | CIFAR-10 | | | | | SVHN | | | |
| Aware | Retrain | 84.15±0.37 | 77.85±1.56 | 86.99±0.78 | 57.42±1.33 | Retrain | 83.70±0.35 | 75.38±1.03 | 93.44±0.88 | 58.58±1.59 |
| | NegGrad | 78.56±0.83 | 68.78±3.12 | 82.56±1.23 | 56.03±0.67 | NegGrad | 81.22±0.58 | 68.89±1.49 | 91.89±1.56 | 57.86±1.23 |
| | Boundary | 55.34±1.58 | 16.99±3.13 | 52.89±3.67 | 60.14±1.34 | Boundary | 65.12±1.56 | 30.01±2.01 | 72.02±1.12 | 87.58±3.91 |
| | SISA | 66.81±0.22 | 53.45±0.68 | 54.49±0.10 | 37.88±0.09 | SISA | 83.01±0.25 | 67.91±0.57 | 82.98±0.96 | 51.03±0.58 |
| | Unrolling | 58.34±1.78 | 31.04±3.52 | 60.34±1.89 | 57.01±1.22 | Unrolling | 70.12±2.13 | 47.23±2.09 | 83.88±0.78 | 55.45±1.23 |
| | T-S | 71.54±1.42 | 71.98±3.12 | 77.01±3.21 | 55.62±1.34 | T-S | 78.52±0.55 | 73.01±0.89 | 91.01±0.78 | 56.12±1.76 |
| | SCRUB | 29.05±1.45 | 0.32±0.10 | 23.89±1.23 | 56.01±2.02 | SCRUB | 23.45±0.10 | 0.23±0.05 | 20.89±0.23 | 64.32±2.33 |
| | DSMixup | 58.19±1.55 | 53.97±1.39 | 63.44±1.22 | 51.12±1.49 | DSMixup | 73.33±0.98 | 65.03±0.87 | 81.48±0.54 | 52.26±1.10 |
| | GLI | 79.19±1.28 | 83.26±1.13 | 84.34±1.08 | 73.37±0.83 | GLI | 82.10±0.83 | 76.08±1.51 | 93.10±0.36 | 78.48±0.36 |
| | LAF+R | 79.20±1.20 | 79.20±0.91 | 84.88±1.21 | 57.84±0.88 | LAF+R | **83.40±0.56** | 76.12±0.68 | 93.68±0.77 | 57.91±0.44 |
| | **Ours** | **82.01±0.68** | **76.97±0.52** | **85.99±1.15** | **57.45±0.71** | **Ours** | 83.37±0.39 | **75.89±0.68** | 93.31±0.55 | 58.11±0.38 |
| Agnostic | LAF | 78.12±1.03 | 73.42±3.56 | 82.33±2.03 | 57.70±0.77 | LAF | 81.60±0.67 | 76.34±1.32 | 92.24±0.67 | 57.80±0.98 |
| | RandLabel | 77.90±2.67 | 72.55±2.88 | 83.38±2.79 | 56.33±0.91 | RandLabel | 77.90±1.24 | 72.55±1.55 | 83.38±0.89 | 56.33±1.33 |
| | L-Mix | 79.01±1.78 | 79.65±2.21 | 84.56±1.46 | 55.89±0.87 | L-Mix | 81.65±0.51 | 75.91±0.97 | 92.44±0.69 | 57.01±1.01 |
| | **Ours** | **79.18±0.98** | **78.48±1.25** | **84.82±1.39** | **57.29±1.02** | **Ours** | **81.77±0.28** | **75.31±1.25** | **92.46±0.47** | **57.88±0.74** |
| | | MNIST | | | | | FASHION-MNIST | | | |
| Aware | Retrain | 99.50±0.08 | 98.83±0.06 | 99.13±0.12 | 49.63±0.64 | Retrain | 96.34±0.49 | 92.34±0.56 | 90.45±0.36 | 47.35±0.86 |
| | NegGrad | 99.02±0.15 | **98.90±0.17** | 98.68±0.35 | 50.67±0.49 | NegGrad | 93.78±0.45 | 89.56±0.35 | 89.44±0.34 | 46.34±0.78 |
| | Boundary | 97.55±1.05 | 94.99±1.69 | 96.05±1.56 | 46.99±2.23 | Boundary | 57.23±3.24 | 46.89±3.04 | 52.09±3.88 | 48.23±1.82 |
| | SISA | 99.11±0.14 | 98.44±0.11 | 99.01±0.08 | 34.89±0.09 | SISA | 91.77±0.30 | 90.89±0.10 | 90.01±0.30 | 31.41±0.19 |
| | Unrolling | 99.64±0.12 | 99.42±0.28 | 99.05±0.24 | 47.89±0.54 | Unrolling | 89.99±0.43 | 84.01±0.94 | 81.43±0.45 | 47.78±0.61 |
| | T-S | 94.35±0.98 | 93.33±2.51 | 93.67±1.23 | 48.01±0.88 | T-S | 83.01±1.34 | 86.89±2.45 | 82.67±1.34 | 44.37±2.40 |
| | SCRUB | 99.17±0.23 | 99.04±0.22 | 98.76±0.12 | 46.88±0.50 | SCRUB | 91.12±0.12 | 88.23±0.45 | 88.88±0.25 | 45.35±0.67 |
| | DSMixup | **99.50±0.14** | 98.51±0.09 | 98.97±0.03 | 47.28±0.05 | DSMixup | 90.85±0.22 | 88.15±0.32 | 88.77±0.31 | 49.92±0.29 |
| | GLI | 99.73±0.07 | 99.70±0.06 | **99.03±0.11** | 39.27±1.35 | GLI | **96.04±0.28** | 96.30±0.91 | **90.46±0.22** | 50.19±0.63 |
| | LAF+R | 99.43±0.16 | 99.20±0.27 | 98.78±0.12 | 49.20±0.57 | LAF+R | 94.23±0.35 | 94.86±1.44 | 90.55±0.35 | 47.42±0.32 |
| | **Ours** | 99.60±0.06 | 98.70±0.28 | 98.60±0.36 | **49.53±0.73** | **Ours** | 95.60±0.86 | **93.01±1.01** | 90.50±0.25 | **47.36±0.22** |
| Agnostic | LAF | 98.22±0.77 | 97.21±1.28 | 97.44±0.81 | 47.79±0.92 | LAF | 91.45±1.46 | 91.01±2.30 | 87.33±2.69 | 46.86±0.76 |
| | RandLabel | 98.51±0.42 | 97.35±0.85 | 97.88±0.51 | 47.81±0.65 | RandLabel | 92.27±1.89 | 93.80±2.53 | 88.75±1.79 | 45.12±1.55 |
| | L-Mix | 99.23±0.07 | 97.56±1.22 | 98.00±0.13 | 45.21±1.01 | L-Mix | 92.58±1.23 | 90.99±2.67 | 88.72±2.44 | 40.89±3.89 |
| | **Ours** | **99.25±0.15** | **97.78±0.98** | **98.13±0.19** | **48.11±0.90** | **Ours** | **92.78±1.18** | **91.99±2.01** | **88.76±1.15** | **46.90±0.69** |

Table 3: Ablation Results for Class-Level Unlearning. MB denotes MixBlock.

| Method | Test$_r$ | Test$_f$ | ASR | Method | Test$_r$ | Test$_f$ | ASR |
|---|---|---|---|---|---|---|---|
| | CIFAR-10 | | | | SVHN | | |
| Retrain | 86.80±0.89 | 0±0 | 67.98±2.21 | Retrain | 94.20±0.78 | 0±0 | 59.63±1.77 |
| *w/o* MB ($\alpha = 0.35$) | 85.01±0.53 | 0±0 | 65.24±0.88 | *w/o* MB ($\alpha = 0.35$) | 92.34±0.69 | 0±0 | 62.87±1.41 |
| *w/o* MB ($\alpha = 0.75$) | 85.42±0.32 | 0±0 | 65.15±0.76 | *w/o* MB ($\alpha = 0.75$) | 92.74±0.49 | 0±0 | 63.24±0.79 |
| *w/o* MB ($\alpha = 1.5$) | 84.96±0.69 | 0±0 | 65.01±0.91 | *w/o* MB ($\alpha = 1.5$) | 92.01±0.86 | 0±0 | 63.17±0.98 |
| *w/o* $L_{real}$ | 29.30±1.80 | 0±0 | 58.42±2.55 | *w/o* $L_{real}$ | 26.89±1.32 | 0±0 | 60.91±1.01 |
| *w/o* $L_{mix}$ | 82.15±1.69 | 0±0 | 61.30±2.31 | *w/o* $L_{mix}$ | 91.68±1.06 | 0±0 | 50.95±2.66 |
| *w/o* Sharpen | 85.83±0.89 | 0±0 | 70.27±0.69 | *w/o* Sharpen | 93.01±0.99 | 0±0 | 67.12±0.68 |
| Ours | 86.32±0.56 | 0±0 | 68.48±0.67 | Ours | 93.40±1.35 | 0±0 | 62.18±0.72 |
| | MNIST | | | | FASHION-MNIST | | |
| Retrain | 98.79±0.20 | 0±0 | 26.56±1.75 | Retrain | 92.71±0.47 | 0±0 | 38.04±2.11 |
| *w/o* MB ($\alpha = 0.35$) | 99.00±0.22 | 0±0 | 30.01±1.02 | *w/o* MB ($\alpha = 0.35$) | 90.96±0.76 | 0±0 | 36.24±1.12 |
| *w/o* MB ($\alpha = 0.75$) | 99.25±0.14 | 0.06±0.08 | 29.37±0.47 | *w/o* MB ($\alpha = 0.75$) | 91.33±0.71 | 0±0 | 36.33±1.09 |
| *w/o* MB ($\alpha = 1.5$) | 99.03±0.18 | 0±0 | 29.01±0.58 | *w/o* MB ($\alpha = 1.5$) | 91.01±0.89 | 0±0 | 36.01±0.98 |
| *w/o* $L_{real}$ | 67.00±1.54 | 1.93±0.13 | 48.63±1.74 | *w/o* $L_{real}$ | 40.50±1.43 | 0±0 | 43.83±1.15 |
| *w/o* $L_{mix}$ | 97.49±0.25 | 0.10±0.06 | 31.30±0.75 | *w/o* $L_{mix}$ | 87.60±0.38 | 0±0 | 38.04±0.62 |
| *w/o* Sharpen | 98.67±0.14 | 0±0 | 30.35±1.19 | *w/o* Sharpen | 91.55±0.36 | 0±0 | 35.24±1.11 |
| Ours | 98.88±0.07 | 0±0 | 27.37±1.60 | Ours | 91.80±1.33 | 0±0 | 38.07±0.65 |

tion without MixBlock (*w/o* MB) yields a Train$_f$ of 95.68, deviating much from the Retrain model. We provide a hyperparameter sensitivity analysis in Appendix A.5.

## 5.6 VISUALIZATION OF REPRESENTATIONS

**Consistency with Retraining**. As shown in Figure 3, MixUnlearn generates a representation space that closely resembles that of the Retrain. The class separation and clustering patterns are visually similar, with the airplane acting as a bridge between the bird, deer, ship, and truck classes. This indicates that MixUnlearn effectively includes the key features of the retrained model. However, this consistency is absent in the initial model, indicating that our method leads to a significant change in the unlearner's behavior.

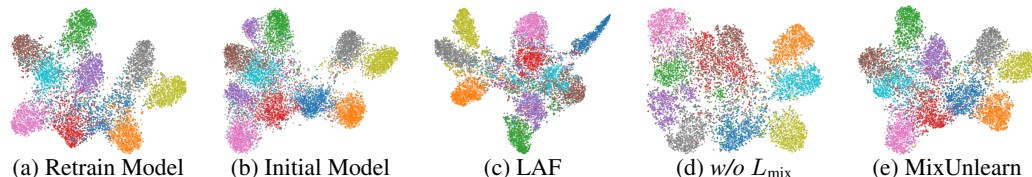

(a) Retrain Model  (b) Initial Model  (c) LAF  (d) *w/o* $L_{\mathrm{mix}}$  (e) MixUnlearn

Figure 3: Representation distributions in the class-level unlearning on the CIFAR-10. The blue denotes forgetting class (class 0; airplane) while the other colors denote the remaining data. The involved classes are: airplane, automobile, bird, cat, deer, dog, frog, horse, ship and truck.

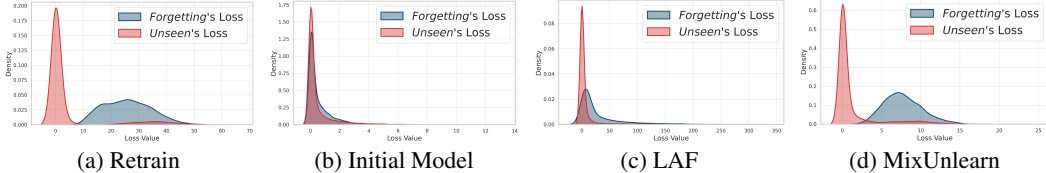

(a) Retrain  (b) Initial Model  (c) LAF  (d) MixUnlearn

Figure 4: Kernel Density Estimate Plots for Loss Distributions. We use the setting of class-level unlearning on CIFAR10. The horizontal axis is CrossEntropyLoss value and the vertical is density.

**Mitigating Catastrophic Unlearning**. The incorporation of $L_{\mathrm{mix}}$ mitigates catastrophic effect. In panel (d), the absence of $L_{\mathrm{mix}}$ leads to more dispersed and less clearly defined clusters, particularly for the bird class. This happens because, when the model forgets the airplane class, it unintentionally also forgets the nearby bird class. A similar catastrophic effect is observed with the state-of-the-art LAF method. However, MixUnlearn successfully mitigates this issue, further supporting the rationale of leveraging mixup to prevent catastrophic unlearning.

## 5.7 KERNEL DENSITY ESTIMATE (KDE) PLOT FOR LOSS DISTRIBUTION

Following Choi et al. (2024) and Chen et al. (2023), we use Kernel Density Estimate (KDE) plots to visualize the distributions of two key cross-entropy losses: Forgetting (blue) and Unseen (red). A successful unlearning is indicated by the visual alignment with the Retrain's loss distributions. Details on plot generation are in Appendix A.7, with results in Figure 4. The Initial Model displays tightly clustered, lower loss values compared to the Retrain Model, while LAF (Figure 4c) shows only a minor shift, suggesting weak unlearning. In contrast, MixUnlearn (Figure 4d) closely matches the loss distributions of the Retrain Model, indicating superior unlearning performance.

## 5.8 ROBUSTNESS CHECK ON NOISY-LABEL OR SEMI-SUPERVISED SETUPS

We evaluate MixUnlearn's robustness in more complex scenarios with noisy or unannotated data, with detailed results provided in Tables 5 and 6 in the Appendix A.6. In the noisy-label setup, it shows resilience, matching the Retrain model in accuracy and Attack Success Rate (ASR), outperforming competitors. In the semi-supervised setting, with unannotated data, it maintains comparable performance. These findings emphasize the robustness of MixUnlearn in addressing both noisy and semi-supervised datasets, making it suitable for real-world applications with imperfect supervision.

## 5.9 EFFICIENCY

We show the time cost comparison in Figure 7 in Appendix A.8. Our method significantly reduces time cost compared to Retrain, the unlearning gold standard, and is faster than advanced models like T-S, SCRUB, and LAF. The efficiency comes from the lightweight MixBlock architecture (66K parameters) and periodic generator updates, making MixUnlearn a fast and effective solution.

## 6 CONCLUSION

In this study, we propose a novel machine unlearning method by leveraging mixup samples to mitigate potential catastrophic unlearning issue. The mixup samples are created in a generator-unlearner framework, where an adversarial generator creates challenging mixup examples that compel the unlearner to draw on information from the forgetting set, while simultaneously losing knowledge from the remaining data. Then unlearner effectively performs unlearning based on these challenging mixed examples. Extensive experiments conducted on benchmark datasets validate the superiority of our method, underscoring the potential of employing mixup techniques for machine unlearning.

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

# A APPENDIX

## A.1 HYPERPARAMETERS AND IMPLEMENTATION

In all experiments, we use a batch size of 32. A learning rate of 1e-5 is applied for CIFAR-10 and SVHN, while 5e-5 is used for MNIST and FASHION-MNIST. The mix ratio $\lambda$ is sampled from beta distribution with $\alpha$ from $\{0.3, 0.5, 0.75, 1, 1.5\}$. $\tau_{\text{gen}}$ is searched from $\{0.05, 0.1, 0.5, 1, 5\}$, $\tau_{\text{mix}}$ is searched from $\{1, 10, 20, 50\}$ and $\tau_{\text{real}}$ is searched from $\{2, 5, 10, 20, 40\}$. The sharpen temperature $T$ is set to 0.3. $\omega$ is tuned from $\{0.5, 1, 10, 20, 30\}$. We implement MixUnlearn in PyTorch, and NVIDIA GeForce RTX 3090 is used for training. For the MNIST and FASHION-MNIST datasets, we set the unlearning epoch to 20. For the CIFAR-10 and SVHN datasets, we set the unlearning epoch to 40.

## A.2 INITIAL MODEL

For the CIFAR-10 and SVHN, we use an 18-layer ResNet (He et al., 2016) architecture, which includes two fully connected layers with output dimensions of 256 and 10. The ResNet model is trained from scratch without using pre-trained weights. For the MNIST and FASHION-MNIST datasets, we employ a CNN architecture (LeCun et al., 1995) consisting of two convolutional layers with 16 and 32 output channels, respectively. The remaining part of the CNN is composed of three fully connected layers with output dimensions of 256, 128, and 10.

To obtain initial models, we train two 18-layer ResNet models on the CIFAR datasets for 20 epochs with a learning rate of 5e-5 and train two CNN models on the MNIST datasets for 10 epochs with a learning rate of 1e-3, following Shen et al. (2024). To ensure model convergence, we examine the learning curves of these initial models, as shown in Figure 5. Notably, we observe that the training accuracy is lower than the testing accuracy for the initial models for CIFAR-10 and SVHN. This can be attributed to the use of data augmentation techniques, such as random cropping, flipping, and normalization, which effectively reduce overfitting by introducing variability during training.

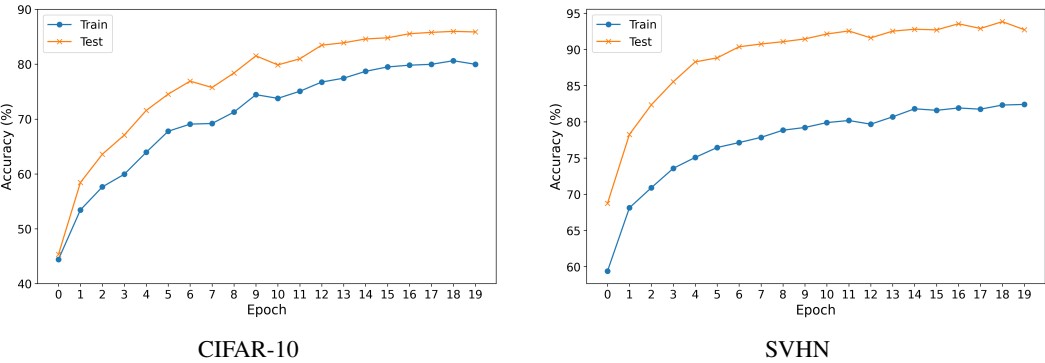

CIFAR-10                    SVHN

Figure 5: Learning Curve of Initial Model.

## A.3 BASELINES

For the gold-standard baseline Retrain, we retrain the CNN models on the MNIST datasets for 20 epochs with a learning rate of 1e-3, while the ResNet models on the CIFAR datasets are retrained for 40 epochs with a learning rate of 5e-5.

For the other baselines, we adhere to the hyperparameters specified in their original studies for the unlearning process, while tuning additional parameters as needed to optimize performance. Neg-Grad updates the deep model's parameters by applying positive gradients to the retaining data and negative gradients to the forgetting data. We adjust the weight for the loss on the retaining data using values from $\{0.001, 0.01, 0.1, 1, 10, 100\}$. Boundary (Chen et al., 2023) modifies the decision boundaries between the forgetting and retaining data to reduce the influence of the forgetting data, with the FSGM bound tuned in the range of 0.1 to 0.5. SISA (Bourtoule et al., 2021) retrains the model on smaller data shards from the remaining dataset and aggregates the results

through ensembling. Unroll (Thudi et al., 2022) employs a regularization technique to minimize verification error, which quantifies the divergence between the unlearned model and the retrained model. For this, we use the default parameters. T-S (Chundawat et al., 2023a) trains two teacher models separately on the forgetting and retaining data, aligning a student model based on differences in the output spaces of the two teachers.[7] We tune the weight for the loss on retaining data from $\{0.0001, 0.01, 0.1, 1, 10, 100\}$ and the temperature from $\{0.5, 1, 2, 3\}$. The student model then achieves unlearning based on the output spaces of these two teachers (each teacher is trained for 10 epochs). SCRUB (Kurmanji et al., 2024) enforces consistency between the model and a teacher model trained on the retaining data while ensuring inconsistency with a teacher model trained on the forgetting data. We tune the weight for the retaining teacher from $\{0.0001, 0.01, 0.1, 1, 10, 100\}$ and the temperature from $\{0.5, 1, 2, 3\}$. DSMixup (Zhou et al., 2022) accelerates SISA by transforming the data shards required for retraining into a smaller set of mixed data shards; we mix two shards in this process. GLI (Choi et al., 2024) perturbs the retaining samples and trains the unlearner on these perturbed samples to maintain generalizability. We tune the perturbation iteration from 2 to 5. For LAF (Shen et al., 2024), we use the best parameters reported in the original paper. For RandLabel, we tune the weight for retaining loss from $\{0.0001, 0.01, 0.1, 1, 10, 100\}$. For L-Mix, we tune alpha (i.e., the parameter from the Beta distribution) from $\{0.3, 0.5, 0.75, 1, 1.5\}$.

**Discussion on Terminating the Unlearning Process**. We adopt the common practice of using a fixed number of unlearning epochs to terminate the unlearning process, as suggested in prior work (Chundawat et al., 2023a; Kurmanji et al., 2024; Choi et al., 2024; Shen et al., 2024). We also track the loss to inspect the progression of the unlearning process. However, determining the optimal strategy for terminating unlearning remains an open question in the field. Unlike traditional learning processes, where validation sets can be used for early stopping, unlearning lacks a clear, standardized approach for effective termination. Should termination rely on metrics such as $Train_r$, $Train_f$, Test, or the Attack Success Rate? Or perhaps a weighted combination of these metrics? For now, we employ the straightforward strategy of setting a fixed unlearning budget to terminate the process. Addressing the question of how to effectively terminate unlearning processes is left as a promising avenue for future research.

## A.4 Ablation Results on Data-Level Unlearning

Table 4: Ablation Results for Data-Level Unlearning (Basic). MB denotes MixBlock.

| CIFAR-10 | | | | | SVHN | | | | |
|---|---|---|---|---|---|---|---|---|---|
| Method | $Train_r$ | $Train_f$ | Test | ASR | Method | $Train_r$ | $Train_f$ | Test | ASR |
| Retrain | 84.15±0.37 | 77.85±1.56 | 86.99±0.78 | 57.42±1.33 | Retrain | 83.70±0.35 | 75.38±1.03 | 93.44±0.88 | 58.58±1.59 |
| w/o MB | 78.86±0.67 | 78.57±0.88 | 84.70±0.96 | 55.19±0.71 | w/o MB | 81.76±0.27 | 76.15±0.64 | 92.06±0.29 | 56.74±0.95 |
| w/o $L_{real}$ | 19.92±1.47 | 0±0 | 16.27±0.76 | 55.52±0.72 | w/o $L_{real}$ | 22.26±1.22 | 0±0 | 21.96±1.45 | 63.35±1.23 |
| w/o $L_{mix}$ | 78.65±0.98 | 80.57±1.11 | 84.62±0.99 | 53.42±0.65 | w/o $L_{mix}$ | 81.61±0.66 | 77.89±0.56 | 92.41±0.96 | 56.32±1.21 |
| w/o Sharpen | 78.97±0.77 | 78.12±0.91 | 84.71±1.11 | 56.99±0.87 | w/o Sharpen | 81.67±0.28 | 75.98±0.94 | 92.33±0.56 | 56.66±1.09 |
| Ours | 79.18±0.98 | 78.48±1.25 | 84.82±1.39 | 57.29±1.02 | Ours | 81.77±0.28 | 75.31±1.25 | 92.46±0.47 | 57.88±0.74 |

| MNIST | | | | | FASHION-MNIST | | | | |
|---|---|---|---|---|---|---|---|---|---|
| Method | $Train_r$ | $Train_f$ | Test | ASR | Method | $Train_r$ | $Train_f$ | Test | ASR |
| Retrain | 99.50±0.08 | 98.83±0.06 | 99.13±0.12 | 49.63±0.64 | Retrain | 96.34±0.49 | 92.34±0.56 | 90.45±0.36 | 47.35±0.86 |
| w/o MB | 98.39±0.41 | 95.62±0.87 | 97.12±0.88 | 46.32±0.54 | w/o MB | 92.54±0.68 | 95.68±1.32 | 88.68±0.97 | 45.99±1.12 |
| w/o $L_{real}$ | 58.16±1.78 | 30.29±1.56 | 49.20±1.33 | 47.89±1.24 | w/o $L_{real}$ | 27.46±1.03 | 4.42±0.41 | 22.44±0.78 | 43.99±0.78 |
| w/o $L_{mix}$ | 99.78±0.08 | 99.86±0.07 | 98.96±0.08 | 48.04±1.01 | w/o $L_{mix}$ | 94.45±0.99 | 96.79±0.95 | 88.56±1.34 | 43.87±1.19 |
| w/o Sharpen | 98.97±0.34 | 97.65±0.94 | 98.01±0.35 | 47.97±0.68 | w/o Sharpen | 92.35±1.01 | 91.86±1.67 | 88.43±0.96 | 46.01±0.76 |
| Ours | 99.25±0.15 | 97.78±0.98 | 98.13±0.19 | 48.11±0.90 | Ours | 92.78±1.18 | 91.99±2.01 | 88.76±1.15 | 46.90±0.69 |

The ablation results in Table 4 show that the full configuration (Ours) consistently outperforms configurations where key components, such as MixBlock (MB), $L_{real}$, and $L_{mix}$, are removed. Specifically, removing MB results in a noticeable performance drop. For example, on the FASHION-MNIST dataset, the full configuration achieves a much closer $Train_f$ to Retrain, compared to the version without MB (91.99% vs. 95.68%), highlighting that MixBlock helps the model forget more effectively. Excluding $L_{real}$ leads to a significant reduction in both training and test accuracy, emphasizing its crucial role in the model's performance. While the removal of $L_{mix}$ also results in lower

---

[7]This baseline is an extended T-S (Chundawat et al., 2023a), which employs trained models as teachers (e.g., partially retrained models) rather than using an initial model alongside a random-parameterized model as teachers.

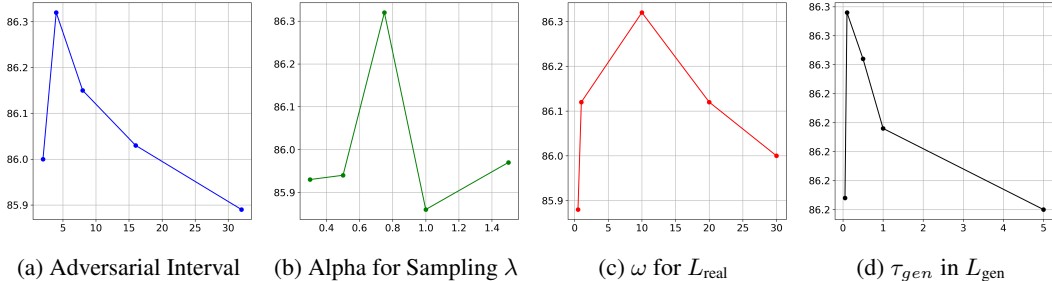

(a) Adversarial Interval    (b) Alpha for Sampling $\lambda$    (c) $\omega$ for $L_{\text{real}}$    (d) $\tau_{gen}$ in $L_{\text{gen}}$

Figure 6: Hyperparameter Sensitivity Analysis on CIFAR-10 class-level unlearning. The X-axis represents the hyperparameter, and the Y-axis shows test accuracy (%) for the remaining classes. The ideal test accuracy for the remaining classes is 87.01% (Retrained model).

performance, its impact is less severe. The Sharpen component has only a marginal effect on overall performance. In summary, these results suggest that $L_{\text{real}}$ is essential for effective unlearning, while both MB and $L_{\text{mix}}$ also contribute positively to performance.

## A.5 SENSITIVITY ANALYSIS ON HYPERPARAMETERS

We perform a hyperparameter sensitivity analysis based on class-level unlearning on CIFAR-10, with the results presented in Figure 6. The hyperparameters analyzed include: 1) the update interval for the generator; 2) the alpha parameter for configuring the Beta distribution ($Beta(\alpha, \alpha)$) used to sample $\lambda$; 3) the loss weight for $L_{\text{real}}$, and 4) the temperature $\tau_{gen}$ in the $SimLoss$ of $L_{\text{gen}}$.

Figure 6 underscores the significance of tuning these parameters to optimize the performance of MixUnlearn. First, the generator update interval must be well-configured: if it is in a small value, the generator may update too quickly, preventing the unlearner from keeping pace. Second, tuning the $\alpha$ parameter is critical. If $\alpha$ is too small, the Beta distribution becomes concentrated at the extremes, overlooking the middle range, which can lead to missed valuable samples. Third, the weight $\omega$ for $L_{\text{real}}$ requires careful balancing. Lastly, fine-tuning the sensitivity of SimLoss, controlled by $\tau_{gen}$, is essential to optimize performance, as it influences temperature scaling during generator training. Despite this, the analysis shows that MixUnlearn is generally robust across a range of values for these hyperparameters, maintaining strong performance.

## A.6 EXPERIMENTS ON NOISY-LABEL OR SEMI-SUPERVISED SCENARIOS

For robustness check, we introduce two additional configurations for evaluation, extending the Data-Level Unlearning (Basic) setup:

**Data-Level Unlearning (Noisy)**. In this setup, we randomly assign incorrect labels to 60% of the training data labeled 0 to 4. These mislabeled samples are then designated as data to be forgotten. The initial model is trained with these noisy labels, while the retrained model is trained with the remaining data (note that remaining data's labels are clean), creating a more complex unlearning environment (Shen et al., 2024). We assess label-aware methods within this noisy context.

**Data-Level Unlearning (Semi-Supervised)**. Here, 60% of the training data remains unannotated. Subsequently, 40% of the samples labeled 5 to 9 are randomly selected for removal. Both the initial and retrained models are trained in a semi-supervised manner using the established Mean-Teacher (Tarvainen & Valpola, 2017)—this introduces complexity in unlearning to approximate a semi-supervisedly retrained model. We evaluate label-agnostic methods in this context.

We show the corresponding results in Table 5 and Table 6. In the noisy-label scenario, our method demonstrates robust performance, as seen in both the accuracy and ASR (Attack Success Rate) metrics which are close to Retrain. This shows that our approach can effectively handle label noise. In the semi-supervised scenario, where a large percentage of the training data remains unannotated, our method continues to show strong results. Specifically, our approach maintains high test accuracy and low ASR, highlighting its ability to deal with missing labels. This is critical in real-world applications, where full supervision is often not feasible. Overall, these results demonstrate that our

Table 5: Unlearning Performance (Mean%±Std%) of Data-Level Unlearning (Noisy). All labels, including potentially noisy ones, are provided to the unlearning algorithm.

| | CIFAR-10 | | | | | SVHN | | | |
|---|---|---|---|---|---|---|---|---|---|
| Method | $Train_r$ | $Train_f$ | Test | ASR | Method | $Train_r$ | $Train_f$ | Test | ASR |
| Retrain | 84.30±0.53 | 3.42±0.35 | 84.69±1.04 | 59.56±0.96 | Retrain | 82.46±0.15 | 2.37±0.23 | 93.38±0.35 | 59.55±1.22 |
| NegGrad | 25.31±2.55 | 5.21±0.83 | 22.14±1.22 | 65.30±3.44 | NegGrad | 18.48±2.68 | 2.48±1.21 | 17.46±6.08 | 58.25±2.05 |
| SISA | 62.31±0.27 | 10.30±0.15 | 53.02±0.63 | 46.23±2.61 | SISA | 80.17±0.13 | 2.45±0.31 | 80.02±0.07 | 44.84±0.04 |
| LAF+R | 77.80±0.89 | 4.88±1.20 | 78.30±0.79 | 59.09±0.88 | LAF+R | 79.39±0.27 | 2.85±0.17 | 90.51±0.50 | **55.09±1.64** |
| **Ours** | **81.12±0.99** | **4.87±1.18** | **82.13±1.04** | **59.14±1.44** | **Ours** | **81.01±0.44** | **2.21±0.37** | **92.18±0.88** | 54.13±0.52 |

Table 6: Unlearning Performance (Mean%±Std%) of Data-Level Unlearning (Semi-Supervised). No label information is exposed to the unlearning algorithm.

| | CIFAR-10 | | | | | SVHN | | | |
|---|---|---|---|---|---|---|---|---|---|
| Method | $Train_r$ | $Train_f$ | Test | ASR | Method | $Train_r$ | $Train_f$ | Test | ASR |
| Retrain | 75.52±0.79 | 73.81±0.71 | 80.72±0.91 | 55.78±1.21 | Retrain | 81.42±1.12 | 70.41±0.69 | 90.50±0.59 | 57.88±0.97 |
| LAF | 65.22±0.75 | 60.03±1.04 | 70.13±0.95 | 52.47±1.05 | LAF | 77.44±0.76 | **71.50±0.43** | 87.04±0.93 | 54.01±1.34 |
| L-Mix | 69.57±1.45 | 62.86±1.13 | 74.65±0.98 | 52.61±1.22 | L-Mix | 77.70±1.09 | 71.73±0.77 | 87.45±1.23 | **54.81±0.89** |
| **Ours** | **71.35±0.89** | **71.76±1.30** | **77.65±1.25** | **55.35±1.89** | **Ours** | **78.58±0.55** | 69.11±0.88 | **89.38±0.66** | 54.50±1.04 |

method is highly adaptable and robust, successfully mitigating the challenges posed by both noisy and semi-supervised datasets, and outperforming other state-of-the-art methods in these settings.

### A.7 PROCEDURE OF VISUALIZING LOSS DISTRIBUTION

To generate KDE plots, we compute two sets of cross-entropy loss values for each method. Forgetting Loss represents the loss on data that has been unlearned, while Unseen Loss corresponds to the loss on entirely new data that the model has not previously encountered. For each method, including Retrain, Initial Model, LAF, and MixUnlearn, we apply the KDE technique to smooth the raw loss values, providing a clearer visualization of the density of loss values. This approach highlights differences in how each method manages forgetting and generalization, offering insights through the visual comparison of loss distributions.

### A.8 TIME COST

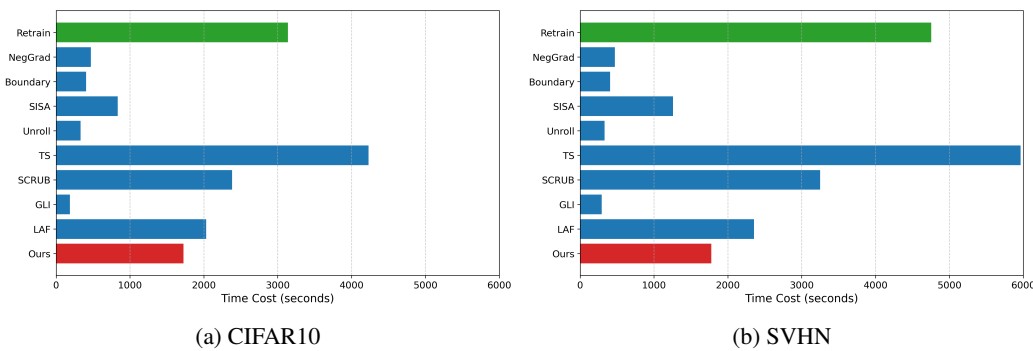

(a) CIFAR10       (b) SVHN

Figure 7: Time cost comparison for class-level unlearning. Experiments are conducted using NVIDIA GeForce RTX 3090. A time cost comparison for class-level unlearning is conducted using an NVIDIA GeForce RTX 3090. SISA utilizes 4 GPU devices to facilitate unlearning, whereas other methods use only a single device.

Compared to Retraining, which requires a complete model retraining and is thus highly time-consuming, our method demonstrates significant superiority in efficiency. This inefficiency highlights the advantages of approximate unlearning techniques. When compared to teacher-student models like T-S and SCRUB, which involve additional computational steps for learning from teacher and student models, our method shows much better time efficiency. Furthermore, our approach outperforms LAF, the current state-of-the-art method based on VAEs with 150K parameters, in both

time efficiency and scalability. With only 66K parameters, our MixBlock architecture provides a lightweight yet highly effective solution, substantially reducing time and resource costs while optimizing the unlearning process. As illustrated in Figure 7, our method consistently achieves low time costs across both CIFAR10 and SVHN datasets.

## A.9 VISUALIZING MIXED SAMPLE

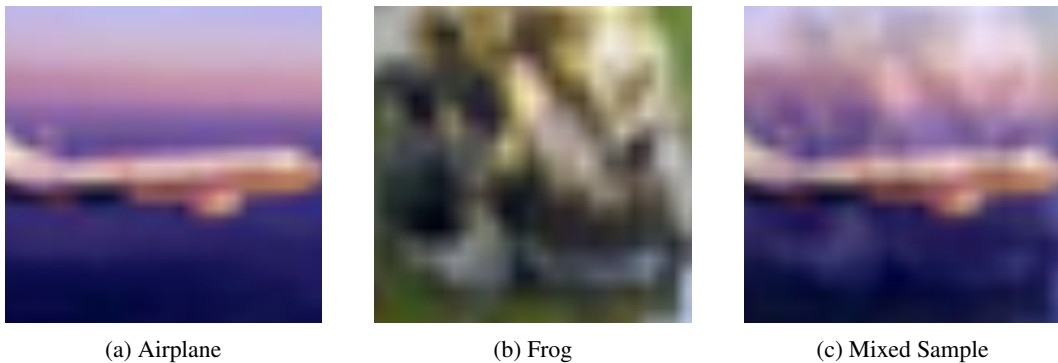

(a) Airplane                    (b) Frog                    (c) Mixed Sample

Figure 8: Visualization of the Mixed Sample. The experiment is conducted on class-level unlearning in CIFAR-10, targeting the unlearning of the 'Airplane' class. $\lambda$ is set as 0.5.

We present the visualization of the mixed sample in Figure 8. From this, we observe that the mixed sample predominantly reveals features of the airplane, while the frog's characteristics are difficult to discern visually. We hypothesize that this image may cause the unlearning model to primarily reveal its shortcomings in forgetting the airplane and failing to effectively retain the frog's information.

## A.10 EFFECT OF DATA RATIO FOR UNLEARNING

We evaluate the effect of the data ratio on our method and baselines by considering different proportions of data removed during Data-Level Unlearning. Specifically, we randomly remove 10% to 40% of the training data labeled with classes 5 through 9 to train the initial model. The results are illustrated in the Figure 9.

To better understand the impact of the data ratio on unlearning performance, we quantify the gap between a focal method (ours or a baseline) and the retrained model. This comparison is performed using two metrics: training accuracy on the forgotten data ($\text{Train}_f$) and testing accuracy. For a given metric, such as $\text{Train}_f$, we calculate the absolute difference between a focal method and the retrained model. This absolute difference represents the gap between the method's performance and the gold standard of retraining. The results of these comparisons are presented in the following figures.

The results demonstrate that increasing the data ratio for forgetting generally makes the unlearning task more challenging, causing larger deviations between unlearning methods and the retrained model. This trend is evident across both $\text{Train}_f$ and Test Accuracy metrics, where baseline methods (e.g., NegGrad, TS, GLI, LAF+R) exhibit increasingly higher gaps as the forgetting data ratio grows. In contrast, our method consistently achieves a low gap relative to the retrained model, maintaining robust performance across all data ratios and datasets (CIFAR-10 and SVHN). This highlights the effectiveness of our approach in handling the unlearning task with minimal deviation, even under more difficult conditions, and underscores its superiority over baseline methods.

## A.11 EXPERIMENTS ON IMAGENET AND VIT

We conduct experiments on the ImageNet dataset (Deng et al., 2009), using the existing ViT model (Dosovitskiy et al., 2021) as the model of interest. Specifically, we utilize the pre-trained weights of "vit-b16-224-in21k", which were trained on ImageNet-21K. For our experiments, we use the validation set of ImageNet-1K, consisting of 50,000 images, as the dataset to manipulate the pre-trained Vision Transformer model for obtaining initial and retrained models. This dataset is further randomly split into a training set of 40,000 images (to construct *Forgetting* and *Remaining* set) and

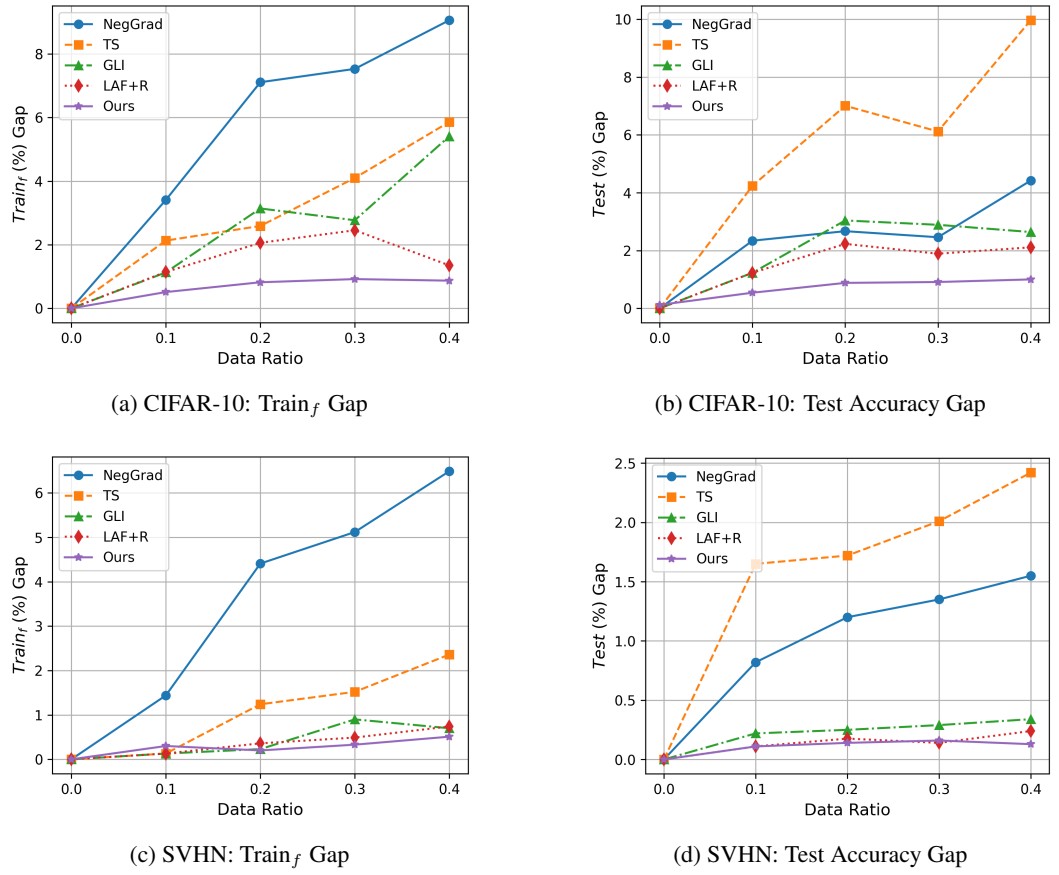

Figure 9: Unlearning gaps between a focal method and the retrained model in terms of $\text{Train}_f$ and Test accuracy.

a testing set of 10,000 images (for validate testing accuracy). For simplicity and clarity, we refer to these subsets as the "training dataset" and "testing dataset" throughout this section.[8]

To obtain the initial and retrained models, we follow a specific procedure. The initial model is trained by fine-tuning "vit-b16-224-in21k" on the 40,000 image training set (20 epochs to ensure convergence). The retrained model is then obtained by training "vit-b16-224-in21k" on a modified version of the training set (25 epochs to ensure convergence), constructed by removing certain data points. Two unlearning scenarios are considered: class-level unlearning, where all data from 100 classes is removed from the training set, and data-level unlearning, where 500 classes are randomly selected and 80% of their data points are removed.

**Quantitative Results**. Table 7 highlights the effectiveness of our proposed method in achieving precise unlearning while preserving critical knowledge. The table presents metrics for both class-level and data-level unlearning scenarios, including $\text{Test}_r$ (test accuracy on remaining classes), $\text{Test}_f$ (test accuracy on forgetting classes), ASR (Attack Success Rate), $\text{Train}_r$ (training accuracy on remaining data), $\text{Train}_f$ (training accuracy on forgotten data) and Test (testing accuracy). Our approach outperforms competing methods across various scenarios. For class-level unlearning, the $\text{Test}_f$ score of $1.51 \pm 0.37$ demonstrates our method's ability to precisely forget targeted knowledge with minimal impact on the remaining data. In the data-level unlearning scenario, our method achieves a $\text{Train}_r$

---

[8]We utilize the ImageNet-1K validation set due to its moderate size and challenging nature, encompassing up to 1,000 classification categories and offering higher resolution than the data used in our main results. While our goal is to evaluate on larger datasets, the computational cost of obtaining initial and retrained models—ensuring convergence for transformer-based architectures and conducting repeated evaluations—renders this impractical. Consequently, we adopt this dataset as a pragmatic choice for evaluation.

Table 7: Unlearning Performance on ImageNet based on ViT.

| | Class-Level Unlearning | | | | Data-Level Unlearning | | | |
|---|---|---|---|---|---|---|---|---|
| Method | Test$_r$ | Test$_f$ | ASR | Method | Train$_r$ | Train$_f$ | Test | ASR |
| Retrain | 78.17±1.88 | 0.00±0.00 | 35.66±2.13 | Retrain | 94.24±1.24 | 58.63±3.12 | 72.58±2.18 | 19.68±2.10 |
| NegGrad | 67.24±1.49 | 2.01±0.42 | 28.46±1.99 | NegGrad | 73.82±2.01 | 28.75±1.45 | 35.97±2.13 | 16.34±1.45 |
| SISA | 70.50±2.09 | 0.00±0.00 | 32.23±1.34 | SISA | 91.41±1.02 | 38.24±2.14 | 68.14±2.56 | 18.24±2.45 |
| T-S | 76.88±1.98 | 25.90±2.45 | 30.46±1.01 | T-S | 92.52±1.34 | 75.77±2.09 | 70.09±1.42 | 16.96±2.34 |
| DSMixup | 68.88±3.02 | 0.00±0.00 | 25.41±2.23 | DSMixup | 89.45±1.86 | 76.14±2.67 | 65.14±1.46 | 17.42±2.34 |
| GLI | 74.78±2.14 | 30.41±3.14 | 31.69±2.31 | GLI | 90.78±1.44 | 69.24±1.64 | 68.14±1.75 | 17.89±1.34 |
| SCRUB | 76.45±0.96 | 24.08±2.97 | 29.06±3.67 | SCRUB | 92.67±1.35 | 76.41±2.13 | 69.31±2.64 | 17.24±2.14 |
| LAF+R | 76.47±1.67 | 23.25±3.98 | 30.18±1.23 | LAF+R | 91.01±1.71 | 87.13±0.97 | 75.64±3.67 | 17.97±1.96 |
| Ours | **76.91±1.84** | **1.51±0.37** | **33.12±1.57** | Ours | **92.89±1.11** | 68.27±2.13 | 74.23±0.75 | 18.98±1.75 |

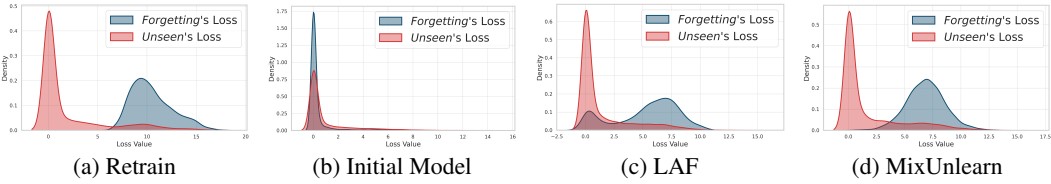

| (a) Retrain | (b) Initial Model | (c) LAF | (d) MixUnlearn |
|---|---|---|---|

Figure 10: Kernel Density Estimate Plots for Loss Distributions. We use the setting of class-level unlearning on ImageNet based on ViT. The horizontal axis is CrossEntropyLoss value and the vertical is density.

of $92.89 \pm 1.11$ and a Test accuracy of $74.23 \pm 0.75$, showcasing its robustness in retaining critical knowledge while effectively handling unlearning tasks.

**Visualization**. The KDE plots in Figure 10 demonstrate that our method closely replicates the behavior of the retrained model compared to the existing state-of-the-art LAF approach. While the initial model shows a significant divergence from the retrained model, our method effectively achieves unlearning and aligns the distribution to better match the retrained model.

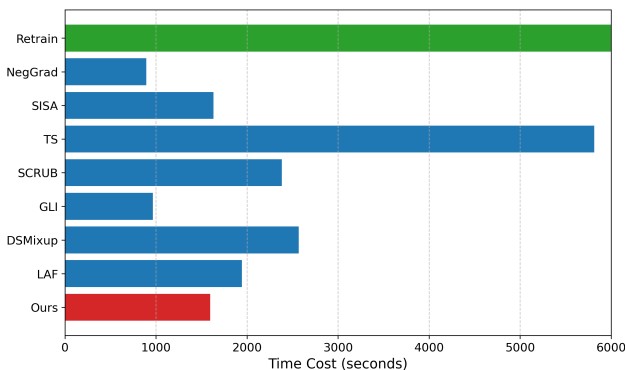

Figure 11: Time cost comparison for class-level unlearning methods on ImageNet using Vision Transformer (ViT). Notably, the SISA approach leverages four GPUs for distributed training. For all methods, we use mixed precision training for acceleration.

**Time Cost Analysis**. Figure 11 highlights the time cost for various unlearning techniques. Approximate unlearning methods demonstrate significant time efficiency by looping over only the forgetting and remaining data pairs, as opposed to retraining, which requires iterations over the entire dataset. Notably, our proposed method achieves superior efficiency compared to other methods, showcasing a substantial reduction in time cost.

## A.12   SHARPEN OPERATION

For the sharpening function, we utilize the standard approach of adjusting the temperature of a categorical distribution, as outlined in Goodfellow et al. (2016). The function is defined as:

$$\text{Sharpen}(q)_i = \frac{q_i^{\frac{1}{T}}}{\sum_{j=1}^{L} q_j^{\frac{1}{T}}},\tag{8}$$

where $q$ represents the input categorical distribution, $i$ indexes a given dimension of $q$, $T$ is the temperature hyperparameter, and $L$ is the total number of categories. As $T \to 0$, the output of $\text{Sharpen}(q)$ converges to a one-hot distribution. In our experiments, we set $T = 0.3$.

