# OpenReview forum: "Adversarial Mixup Unlearning"
_ICLR.cc/2025/Conference — ICLR 2025 Poster_

### Official Review · Reviewer_KJXn · 2024-11-01

**Soundness:** 4
**Presentation:** 4
**Contribution:** 3
**Rating:** 6
**Confidence:** 3

**Summary:**

This paper addresses the challenge of machine unlearning focusing on minimizing the adverse effects of removing data points from forgotten labels. The forgetting process often disrupts the retention of other knowledge, reducing the model’s generalisation, particularly in the intermediate space between forgotten and retained samples.

To mitigate this issue, the authors use data mixup, generating challenging samples through training a generator. These mixed samples reveal forgotten information while disrupting the retention of remaining data, effectively reversing the unlearning process. The unlearner is then trained on these mixup and real samples using contrastive-learning-based losses to improve robustness and efficacy in unlearning.

**Strengths:**

The paper is well-organised, with clear motivations, methodology, and results.

The research addresses a relevant problem and is sound, with a well-motivated toy example that illustrates the proposed approach effectively.

The experimental results provide an adequate comparison to existing methods, that support the method's effectiveness.

**Weaknesses:**

I have no major concerns regarding this work. However, I do have two extended questions:

- I understand that the authors follow experimental setups from previous works on machine unlearning, focusing on low-resolution images and older network architectures. While this is consistent with prior works, these setups may not fully reflect recent advancements in deep learning. I suggest authors to conduct experiments on larger datasets (e.g. ImageNet) and newer model architectures (e.g. ViT, Swin Transformer). I am curious on additional discussion on any computational challenges in scaling up to larger models/datasets and how the method might need to be adapted.

- For challenging mixup samples, the goal is to reveal forgetting samples while lose the remaining samples. In traditional mixup, using a λ close to 1 could align with this aim. I would be interested in understanding how such traditional mixup samples perform within the unlearning process. The authors could consider including an ablation experiment that replaces their adversarial mixup generator with traditional mixup using different fixed λ values (e.g. 0.9, 0.95, 0.99) and compared with their proposed method.

**Questions:**

Please see the Weaknesses

---

> ### Author Response · Authors · 2024-11-23
> **Response to Reviewer KJXn (part 1/2)**
>
> We appreciate the time invested in reviewing this paper, and thank you for the acknowledgment of the paper's novelty.
>
> ---
>
> ## **Q1: More Advanced Dataset and Model**
>
> - **Experiments on ImageNet and ViT:**
>   - Additional experiments were conducted using the **ImageNet dataset** and the **ViT architecture** to evaluate performance under more challenging scenarios, including more classes and higher-resolution images.
>   - These experiments included:
>     - **Quantitative results**
>     - **KDE plot visualizations**
>     - **Comprehensive time-cost analyses**
>   - The findings demonstrated that our model remains **robust** and **highly efficient** even with advanced datasets and models.
>
>   - **Quantitative Results for ImageNet and ViT:**
> | **Class-Level Unlearning**                 |       |       |       | **Data-Level Unlearning**                 |       |       |       |       |
> |--------------------------------------------|-------|-------|-------|-------------------------------------------|-------|-------|-------|-------|
> | **Method**                                 | **Test-r** | **Test-f** | **ASR** | **Method**                                 | **Train-r** | **Train-f** | **Test** | **ASR** |
> | Retrain                                    | 78.17±1.88 | 0.00±0.00 | 35.66±2.13 | Retrain                                    | 94.24±1.24 | 58.63±3.12 | 72.58±2.18 | 19.68±2.10 |
> | NegGrad                                    | 67.24±1.49 | 2.01±0.42 | 28.46±1.99 | NegGrad                                    | 73.82±2.01 | 28.75±1.45 | 35.97±2.13 | 16.34±1.45 |
> | SISA                                       | 70.50±2.09 | 0.00±0.00 | 32.23±1.34 | SISA                                       | 91.41±1.02 | 38.24±2.14 | 68.14±2.56 | 18.24±2.45 |
> | T-S                                        | 76.88±1.98 | 25.90±2.45 | 30.46±1.01 | T-S                                        | 92.52±1.34 | 75.77±2.09 | 70.09±1.42 | 16.96±2.34 |
> | DSMixup                                    | 68.88±3.02 | 0.00±0.00 | 25.41±2.23 | DSMixup                                    | 89.45±1.86 | 76.14±2.67 | 65.14±1.46 | 17.42±2.34 |
> | GLI                                        | 74.78±2.14 | 30.41±3.14 | 31.69±2.31 | GLI                                        | 90.78±1.44 | 69.24±1.64 | 68.14±1.75 | 17.89±1.34 |
> | SCRUB                                      | 76.45±0.96 | 24.08±2.97 | 29.06±3.67 | SCRUB                                      | 92.67±1.35 | 76.41±2.13 | 69.31±2.64 | 17.24±2.14 |
> | LAF+R                                      | 76.47±1.67 | 23.25±3.98 | 30.18±1.23 | LAF+R                                      | 91.01±1.71 | 87.13±0.97 | 75.64±3.67 | 17.97±1.96 |
> | **Ours**                                   | **76.91±1.84** | **1.51±0.37** | **33.12±1.57** | **Ours**                                   | **92.89±1.11** | **68.27±2.13** | **74.23±0.75** | **18.98±1.75** |
>   - Further visualizations and time-cost analyses can be found in **Section A11** of the revised paper.
>
> ---
>
> - **Computational Cost Analysis:**
>   - Our method maintains a **superior rank** while delivering better performance compared to state-of-the-art methods.
>   - Key aspects of computational efficiency:
>     - The number of **outer iterations for unlearning remains fixed**, regardless of model size or image resolution.
>     - Increasing model size or image resolution only impacts **per-image processing time**, not the total iteration count.
>   - This ensures **scalability**, as computational overhead grows primarily with individual image complexity rather than iteration counts. It underscores our model's **scalability** and **efficiency**, securing its **superior rank** among unlearning solutions.

---

> ### Author Response · Authors · 2024-11-23
> **Response to Reviewer KJXn (part 2/2)**
>
> ## Q2: Ablation in Controlling the Mixup Ratio
>
> In the \textit{w/o MB} ablation, we replace the hard mixup samples with vanilla mixup samples and control the mix ratio with different values of $\alpha$:
> - $\alpha=0.35$: Produces mixup ratios skewed close to 1 (minimal mixing).
> - $\alpha=1.5$: Generates mixup ratios centered near 0.5 (equal mixing).
> - $\alpha=0.75$: Produces a broader range of ratios.
>
> Among these, $\alpha=0.75$ consistently yields better results, likely due to its ability to generate diverse mixup samples. Here is the results:
>
> |                    | **CIFAR-10**                           |                    |                    | **SVHN**                              |                    |                    |
> |--------------------|-----------------------------------------|--------------------|--------------------|-----------------------------------------|--------------------|--------------------|
> | **Method**         | **Test_r**                             | **Test_f**         | **ASR**            | **Method**                             | **Test_r**         | **Test_f**         | **ASR**            |
> |--------------------|-----------------------------------------|--------------------|--------------------|-----------------------------------------|--------------------|--------------------|
> | Retrain            | 86.80±0.89                             | 0±0                | 67.98±2.21         | Retrain                                | 94.20±0.78         | 0±0                | 59.63±1.77         |
> | _w/o_ MB (α=0.35)  | 85.01±0.53                             | 0±0                | 65.24±0.88         | _w/o_ MB (α=0.35)                      | 92.34±0.69         | 0±0                | 62.87±1.41         |
> | _w/o_ MB (α=0.75)  | 85.42±0.32                             | 0±0                | 65.15±0.76         | _w/o_ MB (α=0.75)                      | 92.74±0.49         | 0±0                | 63.24±0.79         |
> | _w/o_ MB (α=1.5)   | 84.96±0.69                             | 0±0                | 65.01±0.91         | _w/o_ MB (α=1.5)                       | 92.01±0.86         | 0±0                | 63.17±0.98         |
> | _w/o_ L_real       | 29.30±1.80                             | 0±0                | 58.42±2.55         | _w/o_ L_real                           | 26.89±1.32         | 0±0                | 60.91±1.01         |
> | _w/o_ L_mix        | 82.15±1.69                             | 0±0                | 61.30±2.31         | _w/o_ L_mix                            | 91.68±1.06         | 0±0                | 50.95±2.66         |
> | _w/o_ Sharpen      | 85.83±0.89                             | 0±0                | 70.27±0.69         | _w/o_ Sharpen                          | 93.01±0.99         | 0±0                | 67.12±0.68         |
> | **Ours**           | **86.32±0.56**                         | **0±0**            | **68.48±0.67**     | **Ours**                               | **93.40±1.35**     | **0±0**            | **62.18±0.72**     |
>
> **We are glad to discuss more to address your concerns**.

---

> > ### Comment · Reviewer_KJXn · 2024-11-25
> > **Official Response by Reviewer KJXn**
> >
> > Thank you authors for the rebuttal.
> >
> > I believe it is a good rebuttal.
> > My two extended questions are well addressed.
> > The additional results further strengthen the paper.
> >
> > Given the contributions and the current scoring system, and I find it difficult to justify raising the score to 8.
> > Therefore, I maintain my rating but highly recommend an acceptance.

---

> ### Author Response · Authors · 2024-11-25
> **Thank You!**
>
> Dear Reviewer KJXn,
>
> We sincerely appreciate the time and effort you have invested in reviewing our work. If there are any specific areas where further clarification or additional details would enhance your understanding, please do not hesitate to let us know.
>
> Thank you once again. We wish you all the best!
>
> Sincerely,
>
> The Authors

---

### Official Review · Reviewer_zx2Z · 2024-11-03

**Soundness:** 2
**Presentation:** 3
**Contribution:** 3
**Rating:** 3
**Confidence:** 4

**Summary:**

The author proposes a generator unlarner framework, which uses mixup technology combined with a generator to generate hard samples to help the model retain knowledge about the remaining dataset in forgotten samples. This framework is trained using two contrastive learning loss terms, effectively addressing the issue of potentially losing critical information about retained data during the forgetting process. The author substantiates the effectiveness of the proposed scheme through experimental validation.

**Strengths:**

1. The paper is well-organized and easy to follow, figures and tables are helpful and easy-to-understand. This scheme is relatively simple yet highly effective, as demonstrated by the experimental results, and the designed loss function is particularly noteworthy.
2. The solution proposed by the author applies not only to traditional unlearning scenarios involving labels but also to novel label-agnostic scenarios, demonstrating a broad range of applicability.
3. The author's visualization of the experiment is commendable, providing richer evidence to support the effectiveness of the proposed scheme.

**Weaknesses:**

1. The design of the author's approach does not convincingly demonstrate an actual improvement in unlearning, despite the experimental results resembling retraining. Since retraining does not engage with forgotten data, any generalization enhancements derived from this data will inevitably be lost; otherwise, complete unlearning cannot be assured. The solution proposed by the author incorporates information from forgotten data, which intuitively undermines the concept of complete unlearning.
2. The experiment is not sufficient, and additional deletion results at varying ratios would enhance the credibility. Furthermore, the limited number of datasets and models raises concerns. I am particularly interested in whether the experimental outcomes with more complex models and datasets, such as ImageNet, would influence the results of the method.
3. There is a lack of epoch description of the experiment of the approach, and it remains unclear how many training rounds are necessary for convergence to achieve satisfactory results. The criteria for terminating training are crucial aspects that require discussion in the context of approximate forgetting.
4. The comparative experiments lack sufficient explanation of hyperparameters, and some state-of-the-art results appear somewhat low. I recommend that the author adhere to the hyperparameter guidelines provided in the original paper. While the author states, "if our reproduced results align with the previously reported statistics in Shen et al. (2024), we present their results; otherwise, we provide our results," I believe that directly copying most of the experimental data is unacceptable. The author should revise the baseline results to reflect actual experimental findings. By the way, TS does not require further training of the teacher model; instead, the original and initialized models should serve as the teacher models for the experiment. I suggest the author review the original control scheme rather than relying solely on the introduction in [1]. If I have misunderstood any aspects, I would appreciate clarification from the author. In summary, the author's experimental section appears overly influenced by [1], which raises doubts about the authenticity of the author's experimental workload.

[1] Shaofei Shen, Chenhao Zhang, Yawen Zhao, Alina Bialkowski, Weitong Chen, and Miao Xu. Label-agnostic forgetting: A supervision-free unlearning in deep models. arXiv preprint arXiv:2404.00506, 2024.

**Questions:**

Please check the questions in the weaknesses above.
Additionally, why does the retrained model have a different number of epochs compared to the initial model? Why is the testing accuracy of the model higher than the training accuracy on the CIFAR-10 and SVHN datasets? Does this indicate that the model may not fully converge within the specified number of training rounds, and could certain fine-tuning experimental schemes be affecting the accuracy of the results? Although the author referenced the experimental setup in [1], I think it is essential to ensure that the original model achieves full convergence before drawing any conclusions in research related to unlearning.

---

> ### Author Response · Authors · 2024-11-23
> **[ We are open to discuss! ] Response to zx2Z (Part 1/5 )**
>
> We thank you for your comments and interest in our work. We have taken your suggestion into account when revising our paper. However, we must clarify some misunderstandings that may affect the scoring of our paper.
>
>
> ## Q1: Can our method assure precise unlearning? (We avoid using "complete unlearning" as we cannot clearly define its meaning and it is not a widely-used term)
>
> There is a misunderstanding here. Let us clarify two points:
>
> 1. **Our method does not violate the principles of approximate unlearning**
>    Our paper focuses on *approximate unlearning*, which aims to forget data from a well-trained model. Leveraging forgetting information to facilitate unlearning is a common practice in this domain (leveraging forgetting data does not necessarily mean embedding them into the model). For example, approximate unlearning methods like NegGrad, Boundary, and T-S all require the forgetting data as input. Therefore, our method does not violate the principles of approximate unlearning.
>
> 2. **MixUnlearn advances approximate unlearning towards more precise unlearning**
>    Contrary to your point about "undermining complete unlearning", MixUnlearn enhances approximate unlearning by achieving a higher degree of precision.
>
>    **Let's consider the example from our motivation: What would an ideal solution look like for unseen data?**
>
> For an unseen sample, the ideal outcome is for it to retain only the knowledge derived from the remaining data while forgetting what it should forget from the removed (or forgetting) data. This aligns with the notion of "complete unlearning" you referenced.
>    The goal of MixUnlearn is precisely this: ensuring that an unseen sample forgets information tied to forgetting data while preserving generalizability based on the remaining data. Please refer to **Figure 1** (where unseen samples **only** benefit from the remaining data) and **Eq. 5** for a clearer understanding of the aim and mechanisms of MixUnlearn.

---

> ### Author Response · Authors · 2024-11-23
> **[ We are open to discuss! ] Response to zx2Z (Part 2/5)**
>
> ## Q2: The Experiments Are Not Sufficient? Now We Have More Evaluation Results.
>
> Thank you for your constructive suggestions. We have incorporated more thorough experiments to evaluate our method. Specifically, we have:
>
> - **Expanded data ratio setups:** We evaluated our method under varying data ratio setups. The takeaway is that our method is robust across these setups. Appendix A 10 presents a series of experiments about data ratio.
>
> - **Experiments on ImageNet and ViT:**
>   - Additional experiments were conducted using the **ImageNet dataset** and the **ViT architecture** to evaluate performance under more challenging scenarios, including more classes and higher-resolution images.
>   - These experiments included:
>     - **Quantitative results**
>     - **KDE plot visualizations**
>     - **Comprehensive time-cost analyses**
>   - The findings demonstrated that our model remains **robust** and **highly efficient** even with advanced datasets and models.
>
> ### Quantitative Results for ImageNet with ViT
>
> | **Class-Level Unlearning**                 |       |       |       | **Data-Level Unlearning**                 |       |       |       |       |
> |--------------------------------------------|-------|-------|-------|-------------------------------------------|-------|-------|-------|-------|
> | **Method**                                 | **Test-r** | **Test-f** | **ASR** | **Method**                                 | **Train-r** | **Train-f** | **Test** | **ASR** |
> | Retrain                                    | 78.17±1.88 | 0.00±0.00 | 35.66±2.13 | Retrain                                    | 94.24±1.24 | 58.63±3.12 | 72.58±2.18 | 19.68±2.10 |
> | NegGrad                                    | 67.24±1.49 | 2.01±0.42 | 28.46±1.99 | NegGrad                                    | 73.82±2.01 | 28.75±1.45 | 35.97±2.13 | 16.34±1.45 |
> | SISA                                       | 70.50±2.09 | 0.00±0.00 | 32.23±1.34 | SISA                                       | 91.41±1.02 | 38.24±2.14 | 68.14±2.56 | 18.24±2.45 |
> | T-S                                        | 76.88±1.98 | 25.90±2.45 | 30.46±1.01 | T-S                                        | 92.52±1.34 | 75.77±2.09 | 70.09±1.42 | 16.96±2.34 |
> | DSMixup                                    | 68.88±3.02 | 0.00±0.00 | 25.41±2.23 | DSMixup                                    | 89.45±1.86 | 76.14±2.67 | 65.14±1.46 | 17.42±2.34 |
> | GLI                                        | 74.78±2.14 | 30.41±3.14 | 31.69±2.31 | GLI                                        | 90.78±1.44 | 69.24±1.64 | 68.14±1.75 | 17.89±1.34 |
> | SCRUB                                      | 76.45±0.96 | 24.08±2.97 | 29.06±3.67 | SCRUB                                      | 92.67±1.35 | 76.41±2.13 | 69.31±2.64 | 17.24±2.14 |
> | LAF+R                                      | 76.47±1.67 | 23.25±3.98 | 30.18±1.23 | LAF+R                                      | 91.01±1.71 | 87.13±0.97 | 75.64±3.67 | 17.97±1.96 |
> | **Ours**                                   | **76.91±1.84** | **1.51±0.37** | **33.12±1.57** | **Ours**                                   | **92.89±1.11** | **68.27±2.13** | **74.23±0.75** | **18.98±1.75** |
>
> Further **visualization** and **time-cost analyses** can be found in Section A11 of the revised paper.

---

> ### Author Response · Authors · 2024-11-23
> **[ We are open to discuss! ] Response to zx2Z (Part 3/5)**
>
> ### Q3: Description of Epochs for Unlearning? How to Terminate Unlearning?
>
> - We have included the number of unlearning epoch in the revised paper. Thank you for your suggestion.
>
> - **How to terminate unlearning** remains an open question as future research, and is out of the scope of our work. In our work, we employ the common practice [1][2][3] of terminating the unlearning process with a fixed epoch number. Each method is allocated the same unlearning epoch budget, and we verify that the training loss has converged during this process. In our experiments, baseline methods are allocated a total of 40 unlearning epochs. Additionally, we monitor the training loss to track the progression of unlearning.
>
>   Determining an optimal strategy for terminating unlearning is an unresolved challenge in the field. Unlike traditional learning processes, which often rely on validation sets for early stopping, unlearning lacks a standardized approach for effective termination. Key questions include:
>   - Should termination depend on metrics like training loss before unlearning (\(Train_r\)), training loss after unlearning (\(Train_f\)), test loss, or the attack success rate?
>   - Could a weighted combination of these metrics provide a more robust criterion?
>
>   For now, we adopt the straightforward approach of setting a fixed unlearning epoch budget. Addressing the broader question of effective termination strategies in unlearning processes remains a promising avenue for future research. **The discussion for criteria for terminating has been included in Appendix A3**.
>
>   **Note**: We do monitor the model's loss curve as a basic manner to ensure convergence during unlearning.
>
> ---
>
> #### References
> [1] Chundawat, V. S., Tarun, A. K., Mandal, M., et al. *Can bad teaching induce forgetting? Unlearning in deep networks using an incompetent teacher*. Proceedings of the AAAI Conference on Artificial Intelligence, 2023, 37(6): 7210-7217.
>
> [2] Choi, D., Choi, S., Lee, E., et al. *Towards Efficient Machine Unlearning with Data Augmentation: Guided Loss-Increasing (GLI) to Prevent the Catastrophic Model Utility Drop*. Proceedings of the IEEE/CVF Conference on Computer Vision and Pattern Recognition, 2024: 93-102.
>
> [3] Shen, S., et al. *Label-Agnostic Forgetting: A Supervision-Free Unlearning in Deep Models*. The Twelfth International Conference on Learning Representations.

---

> ### Author Response · Authors · 2024-11-24
> **[ We are open to discuss! ] Response to zx2Z (Part 4/5)**
>
> ### Q4: Lack sufficient explanation of hyperparameters? Experimental workload is low? The setup of the T-S model?
>
> - **Reproduced Results**:
>   We have utilized our reproduced results for evaluation. Significant efforts were made to ensure critical evaluation of the open benchmark. Before the initial submission, we reproduced all benchmark results rather than solely relying on cited experimental results from the existing benchmarking in the ICLR 24' paper [3]. In the revised paper, we now use all our reproduced results (which only affect some baselines but retain similar findings). These reproduced results closely match open benchmarks, except under noisy label conditions, where we provided specific findings. The steps taken during the **initial submission** and **this revision** include:
>   1. Reproducing open benchmark results with thorough parameter tuning before the initial submission.
>   2. Reporting results from reproduced noisy label setups, noting observed differences.
>   3. Introducing stronger baselines, such as RandLabel and L-Mix, for robust comparisons.
>   4. Extending evaluations to include methods like DSMixup and GLI.
>   5. Incorporating additional evaluation settings, such as semi-supervised scenarios.
>
>   Full details of the parameter search for baselines and reproduced results are available in **Section 5.4**.
>
> - **Setup of T-S Model (Note that We use Extended One)**:
>   The original T-S paper [1] has discussed using an additional trained model as a teacher (e.g., using a partially re-trained model as a teacher). The ICLR 24' paper [3] extends this concept by using a trained model to explore forgetting and retention. We tested both the simplest setup of T-S and the advanced setup in the ICLR 24' paper. Our findings show that the latter performs better, and we aligned our experiments accordingly. The revised paper now includes more detailed discussions about this extension. Additionally, we are willing to release our implementations for clarity.
>
> - **Code Release**:
>   We are glad to release our code to ensure transparency in evaluation.
>
> #### Reference:
> [1] Chundawat V S, Tarun A K, Mandal M, et al. Can bad teaching induce forgetting? unlearning in deep networks using an incompetent teacher[C]//Proceedings of the AAAI Conference on Artificial Intelligence. 2023, 37(6): 7210-7217.
>
> [3] Shen, Shaofei, et al. "Label-Agnostic Forgetting: A Supervision-Free Unlearning in Deep Models." The Twelfth International Conference on Learning Representations.

---

> ### Author Response · Authors · 2024-11-24
> **[ We are open to discuss! ] Response to zx2Z (Part 5/5)**
>
> ### Q5: Convergence of the Model? Why is the Testing Accuracy of the Model Higher Than the Training Accuracy?
>
> These questions are centered around model convergence. Below are our explanations and clarifications:
>
> 1. We prove **Model Convergence**
>    - We assign different numbers of epochs to the initial model and the retrained model to ensure convergence. Our hypothesis is that the initial model, having more data points for gradient calculation, converges more robustly and faster. To account for this, we assign fewer epochs to the initial model.
>    - The corresponding learning curves are included in **Appendices A2** and **A3**.
>
> 2. **Testing Accuracy Higher Than Training Accuracy** is possible
>    - Observing higher testing accuracy compared to training accuracy is not uncommon, especially when data augmentation and regularization methods (e.g., dropout) are applied.
>    - In our case, we use data augmentation for CIFAR and SVHN datasets, which prevents overfitting to the training data. Consequently, the testing accuracy is consistently higher than the training accuracy.
>    - This explanation is included in **Appendix A2**. Additional evidence supporting this phenomenon can be found in the following references:
>      - [Official Reply in Google Research](https://github.com/google-research/vision_transformer/issues/30)
>      - [Stack Overflow Discussion](https://stackoverflow.com/questions/42878683/keras-cifar10-example-validation-and-test-loss-lower-than-training-loss)
>    - To further validate, we tested the impact of removing data augmentation, which resulted in training accuracy exceeding testing accuracy but poorer overall model performance.
>
> 3. **Regarding the Question on Convergence and Fine-Tuning**
>    - We have demonstrated model convergence; hence, this is no longer a concern. All unlearning methods begin with a "well-trained" model, ensuring a fair starting point.
>
> ---
>
> **We are confident in addressing your comments and are happy to discuss further to resolve any additional concerns.**

---

> ### Author Response · Authors · 2024-11-26
> **We Truly Value Your Feedback: We Need Further Discussion**
>
> Dear Reviewer zx2Z,
>
> We sincerely appreciate your insightful comments on MixUnlearn and are grateful for the opportunity to address your concerns. We understand that our rebuttal may require some of your valuable time, so we have provided a brief summary of the key points addressed in our response:
>
> 1. **Clarification of MixUnlearn’s Contribution:** We illustrate that *MixUnlearn* effectively advances approximate unlearning toward a more precise unlearning solution.
> 2. **Additional Experiments:** We have added data ratio experiments, as well as new experiments on ImageNet using the Vision Transformer (ViT), including not only quantitative results but also visualizations and efficiency analysis.
> 3. **Unlearning Termination Process:** We clarify the methodology used to determine when unlearning is complete.
> 4. **Reproducibility of Results:** We have revisited our results back to our reproduced ones, and clarified how the T-S baseline was set up in our experiments.
> 5. **Convergence Evidence:** We provide additional evidence demonstrating the convergence behavior of both the initial and retrained models.
>
> We hope these updates address your previous concerns effectively. Please let us know if any points still require further clarification or if you would like more detailed information.
>
> Thank you once again for your thoughtful feedback.
>
> Kind regards,
> The Authors

---

> > ### Comment · Area_Chair_yWsg · 2024-12-01
> >
> > Hi Reviewer zx2Z,
> >
> > The authors have provided new results and responses - do have a look and engage with them in a discussion to clarify any remaining issues as the discussion period is coming to a close in less than a day (2nd Dec AoE for reviewer responses).
> >
> > Thanks for your service to ICLR 2025.
> >
> > Best,
> > AC

---

> ### Author Response · Authors · 2024-12-02
> **Our Final Effort to Address Misunderstandings**
>
> As we approach the final day of the discussion phase, we would like to make one last effort to address the misunderstandings raised regarding our paper. Below, we provide clarifications on several key points:
>
> - **Clarification on the Common Practice of Unlearning**:
>   The incorporation of "forget" data to facilitate unlearning (without embedding them into the model), such as in the well-established NegGrad method, is a standard practice. Our method does **not** violate the principles of unlearning. In fact, it achieves more precise unlearning by ensuring that unseen samples retain generalizability from the remaining data, while **not** benefiting from the "forgotten" data.
>
> - **Clarification on Model Convergence**:
>   The initial model used in our evaluation was validated for convergence. The observation that "test accuracy exceeds train accuracy" is a common phenomenon in datasets such as CIFAR-10 and SVHN.
>
> - **Clarification on Workload**:
>   We have provided reproduced results from our own experiments. Our intention was to avoid any potential discrepancies between our benchmarking results and previous ones, although the differences are minimal. Additionally, the use of T-S is simply an extension of the methodology discussed in the original paper.
>
> - **Clarification on Unlearning Termination**:
>   We adhere to standard practices for unlearning progression. However, we also introduce a new discussion on the termination of unlearning. This is still an open question, and while relevant, it lies outside the scope of our paper.
>
> - **Incorporation of Additional Experiments**:
>   1. We present results demonstrating the performance of our method across different data ratios.
>   2. We also provide new experiments on more complex datasets and models, such as ImageNet and Vision Transformers (ViT).
>
> We still want your valuable response for this rebuttal. We hope these clarifications help you reconsider the contributions and validity of our paper. We sincerely appreciate the time and effort you have dedicated to reviewing our work.

---

### Official Review · Reviewer_MH1a · 2024-11-03

**Soundness:** 3
**Presentation:** 2
**Contribution:** 3
**Rating:** 6
**Confidence:** 2

**Summary:**

The paper introduces a novel and unique generator-unlearner method to yield mixed examples, which helps improve  machine unlearning effectiveness. Especially , this paper use a  novel contrastive objective not only to help refine generator, but also help unlearner  retain knowledge of remaining data and forget  forgetting data.  Hence, this operation can   prevent  catastrophic unlearning issue.  Furthermore, extensive experiments  demonstrate the efficiency and effectiveness of their proposed method.

**Strengths:**

1. This paper uses a intriguing adversarial process to solve the maching unlearning issue. They use a unique generator-unlearner adversarial process to force unlearner to  forget the forgetting and retain the remaining. This proposed method are novel and effective.
2. The proposed method outperforms previous work in a variety of settings, including both label-agnostic and label-aware settings, and has a relatively small time cost.

**Weaknesses:**

1. Though their experiments are relatively comprehensive,  the underlying dataset are somehow small. More experimental results on large dataset are expected to be demonstrated (e.g. imagenet).

2. The presentation of the paper needs to be improved, especially the description of the methods section.

**Questions:**

Why MixUnlearn sometimes demonstrates better performance than retrain (e.g., on CIFAR-10)?

---

> ### Author Response · Authors · 2024-11-23
> **Response to Reviewer MH1a**
>
> Thank you for your appreciation in the novelty our MixUnlearn. We deeply appreciate your constructive feedback, which has helped us enhance the quality of our work.
>
> ## Q1: More Advanced Dataset and Model
>
> We conducted additional experiments using the ImageNet dataset and the ViT architecture, targeting more challenging scenarios with more classes and higher image resolutions. These experiments include **quantitative results**, **KDE plot visualizations**, and comprehensive **time-cost analyses**. The findings demonstrate that our model remains robust and highly efficient, even with a more advanced model and a challenging dataset.
>
> ### Quantitative Results
>
> | **Class-Level Unlearning**                 |       |       |       | **Data-Level Unlearning**                 |       |       |       |       |
> |--------------------------------------------|-------|-------|-------|-------------------------------------------|-------|-------|-------|-------|
> | **Method**                                 | **Test-r** | **Test-f** | **ASR** | **Method**                                 | **Train-r** | **Train-f** | **Test** | **ASR** |
> | Retrain                                    | 78.17±1.88 | 0.00±0.00 | 35.66±2.13 | Retrain                                    | 94.24±1.24 | 58.63±3.12 | 72.58±2.18 | 19.68±2.10 |
> | NegGrad                                    | 67.24±1.49 | 2.01±0.42 | 28.46±1.99 | NegGrad                                    | 73.82±2.01 | 28.75±1.45 | 35.97±2.13 | 16.34±1.45 |
> | SISA                                       | 70.50±2.09 | 0.00±0.00 | 32.23±1.34 | SISA                                       | 91.41±1.02 | 38.24±2.14 | 68.14±2.56 | 18.24±2.45 |
> | T-S                                        | 76.88±1.98 | 25.90±2.45 | 30.46±1.01 | T-S                                        | 92.52±1.34 | 75.77±2.09 | 70.09±1.42 | 16.96±2.34 |
> | DSMixup                                    | 68.88±3.02 | 0.00±0.00 | 25.41±2.23 | DSMixup                                    | 89.45±1.86 | 76.14±2.67 | 65.14±1.46 | 17.42±2.34 |
> | GLI                                        | 74.78±2.14 | 30.41±3.14 | 31.69±2.31 | GLI                                        | 90.78±1.44 | 69.24±1.64 | 68.14±1.75 | 17.89±1.34 |
> | SCRUB                                      | 76.45±0.96 | 24.08±2.97 | 29.06±3.67 | SCRUB                                      | 92.67±1.35 | 76.41±2.13 | 69.31±2.64 | 17.24±2.14 |
> | LAF+R                                      | 76.47±1.67 | 23.25±3.98 | 30.18±1.23 | LAF+R                                      | 91.01±1.71 | 87.13±0.97 | 75.64±3.67 | 17.97±1.96 |
> | **Ours**                                   | **76.91±1.84** | **1.51±0.37** | **33.12±1.57** | **Ours**                                   | **92.89±1.11** | **68.27±2.13** | **74.23±0.75** | **18.98±1.75** |
>
> We recommend referring to Section A11 in the revised paper for further **visualisation and time cost analysis**.
>
> ---
>
> ## Q2: Why MixUnlearn Sometimes Demonstrates Better Performance than Retrain (e.g., on CIFAR-10)?
>
> This arises from the fact that the unlearning algorithm starts from a well-trained model with better generalizability compared to the retrained model. For instance, a well-trained CIFAR-10 classifier typically outperforms a retrained model trained on a smaller dataset (due to the removal of forgetting data).
>
> Considering the transition from high to low generalizability during unlearning, the algorithm may retain better generalizability than a retrained model due to imperfect forgetting. This underpins our claim that "Close to Retrain is Better." Importantly, our model exhibits behavior that is closest to retraining among existing solutions.
>
> **We are glad to provide more results to address your concerns**.

---

> > ### Comment · Reviewer_MH1a · 2024-11-25
> > **Official Response**
> >
> > Thank you for your rebuttal.
> >
> >
> > My main concerns have been solved. I believe 6 is my final score.

---

> > > ### Author Response · Authors · 2024-11-25
> > > **Thank You!**
> > >
> > > Dear Reviewer MH1a,
> > >
> > > We deeply appreciate the time and effort you have dedicated to reviewing our work, as well as your prompt response. Thank you once again, and we wish you all the best!
> > >
> > > The Authors

---

### Author Response · Authors · 2024-11-23
**Summary of Revision**

**Summary of Revision**

We sincerely thank the reviewers for their detailed review, valuable feedback, and acknowledgment of the novelty of our work. To address the concerns raised and enhance the quality of our paper, we have made the following revisions:

1. **Expanded Experiments on Challenging Datasets**
   - We conducted additional experiments using the ImageNet dataset and the ViT architecture, targeting more challenging scenarios. These include quantitative results, KDE plot visualizations, and comprehensive time-cost analyses.
   - Details can be found in **Appendix A11**.

2. **Evaluation of Robustness Across Data Sizes**
   - We benchmarked our method's performance across varying dataset sizes to demonstrate its robustness, regardless of the forgetting dataset size.
   - The results are presented in **Appendix A10**.

3. **Reproduction of Benchmark Results (Clarifying No Differences in Findings)**
   - Before first submission, we have reproduced all benchmark results rather than relying solely on cited experiment results from existing benchmarking in ICLR 24' paper [1]. Now, we use all our reproduced results in revised paper (only affecting some baselines but remaining similar findings). Our reproduced results closely match open benchmarks, except under noisy label conditions, where we provided our specific findings. Steps taken in **first submission** and **this revision** include:
     - Reproducing open benchmark results with thorough parameter tuning before the initial submission.
     - Reporting results from reproduced noisy label setups, noting differences.
     - Introducing stronger baselines, such as **RandLabel** and **L-Mix**, for robust comparisons.
     - Extending evaluations to include methods like **DSMixup** and **GLI**.
     - Incorporating additional evaluation settings, such as semi-supervised scenarios.
   - Full details and reproduced results are available in **Section 5.4**.

4. **Demonstrating Model Convergence**
   - To provide evidence of model convergence, we added learning curves for our models.
   - These are included in **Appendices A2 and A3**.

5. **Comprehensive Ablation Studies**
   - We conducted ablation studies to analyze the effect of the mixup ratio, offering deeper insights into its influence on model performance.

We remain open to providing further results or clarifications to address any additional concerns. Thank you once again for your thoughtful review and constructive feedback.

[1] Shen, Shaofei, et al. "Label-Agnostic Forgetting: A Supervision-Free Unlearning in Deep Models." The Twelfth International Conference on Learning Representations.

---

### Author Response · Authors · 2024-11-25
**Kind Reminder: Seeking Reviewer Feedback for Author/Reviewer Discussion Phase**

Dear Reviewers,

We sincerely appreciate your invaluable feedback, which has been instrumental in improving our submission. In response to your reviews, we have thoughtfully addressed each of your comments and hope our revisions meet your expectations.

As the author/reviewer discussion phase (Nov 12–26) deadline is approaching, we kindly seek your input on our responses. We understand your time is limited and greatly value your insights. Engaging in further discussion would be immensely helpful in ensuring our submission aligns with the high standards of ICLR 2025.

Thank you again for your time and consideration. We are eager to collaborate further and refine our work.

Best regards,

The Authors

---

### Meta-Review · Area_Chair_yWsg · 2024-12-26

**Metareview:**

The paper presents an algorithm for machine unlearning, focusing on the interface between the forget set and remain set. A mixup-based generator in combination with contrastive losses are used to sharpen the model's decision boundary between the forget and remain set by incorporating hard synthetic samples into the unlearning process. Experiments on a range of standard unlearning datasets with the addition of ImageNet show the effectiveness of the method.

The paper introduces a novel adversarial process and contrastive losses and extensive evaluation on multiple datasets including ImageNet that validate the effectiveness of the approach. The method also applies to label-agnostic unlearning in addition to label-aware unlearning. Overall it presents a good empirical study.

**Additional Comments On Reviewer Discussion:**

There were initial concerns from multiple reviewers about the sufficiency of the experiments, particularly the performance on more challenging datasets, robustness to different forgetting dataset sizes, and comparison to a vanilla mixup baseline. There were also concerns on the convergence and reproducibility of results. In the rebuttal, the authors included substantial additional results on ImageNet and ViT architectures, and ablation studies on varying dataset sizes and mixup ratios. These additional results addressed the concerns of reviewers KJXn and MH1a to their satisfaction (though they did not raise their score, KJXn highly recommends acceptance), while reviewer zx2Z did not respond to the rebuttal; the AC has taken a look and agrees with the authors that the concerns of zx2Z are addressed.

---

### Decision · Program_Chairs · 2025-01-22

Accept (Poster)